# Enhancing Sample Selection Against Label Noise by Cutting Mislabeled Easy Examples

**Suqin Yuan[1]**    **Lei Feng[2]***    **Bo Han[3]**    **Tongliang Liu[1]***
[1] Sydney AI Centre, The University of Sydney
[2] Southeast University [3] Hong Kong Baptist University

## Abstract

Sample selection is a prevalent approach in learning with noisy labels, aiming to identify confident samples for training. Although existing sample selection methods have achieved decent results by reducing the noise rate of the selected subset, they often overlook that not all mislabeled examples harm the model's performance equally. In this paper, we demonstrate that mislabeled examples correctly predicted by the model early in the training process are particularly harmful to model performance. We refer to these examples as *Mislabeled Easy Examples* (MEEs). To address this, we propose *Early Cutting*, which introduces a recalibration step that employs the model's later training state to re-select the confident subset identified early in training, thereby avoiding misleading confidence from early learning and effectively filtering out MEEs. Experiments on the *CIFAR*, *WebVision*, and full *ImageNet-1k* datasets demonstrate that our method effectively improves sample selection and model performance by reducing MEEs. Our implementation can be found at https://github.com/tmllab/2025_NeurIPS_MEE.

## 1 Introduction

Deep Neural Networks (DNNs) have achieved remarkable success, while heavily relying on the availability of high-quality, accurately annotated data. In practice, collecting large-scale datasets with precise labels is challenging due to the high costs involved and the inherent subjectivity of manual annotation processes. Consequently, datasets often contain noisy labels, which can degrade the generalization performance of DNNs—a problem known as learning with noisy labels (LNL) (Natarajan et al., 2013). One prevalent approach to address LNL is sample selection, which aims to identify confident samples for training while discarding potentially mislabeled ones.

Sample selection methods can be categorized into two types: loss-based and dynamics-based. Loss-based methods rely on the assumption that clean samples tend to have smaller loss values than mislabeled samples (Han et al., 2018; Liu et al., 2020; Xia et al., 2021; Li et al., 2024). In contrast, dynamics-based methods exploit the memorization effect of DNNs, which suggests that DNNs learn simple patterns first and then gradually fit the assigned label for each particular minority instance, including mislabeled samples (Liu et al., 2020; Zhang et al., 2021; Yuan et al., 2024). By analyzing the learning dynamics of DNNs, these methods aim to identify clean samples that are learned early and consistently throughout the training process (Yu et al., 2019; Xia et al., 2020a; Bai and Liu, 2021; Wei et al., 2022), considering them as confident samples for training. In recent years, dynamics-based methods have gained attention due to their ability to select Clean Hard Examples (CHEs)—challenging clean samples that are difficult to identify but crucial for achieving near-optimal generalization performance (Feldman and Zhang, 2020; Bai and Liu, 2021; Yuan et al., 2023).

---

*Corresponding authors.

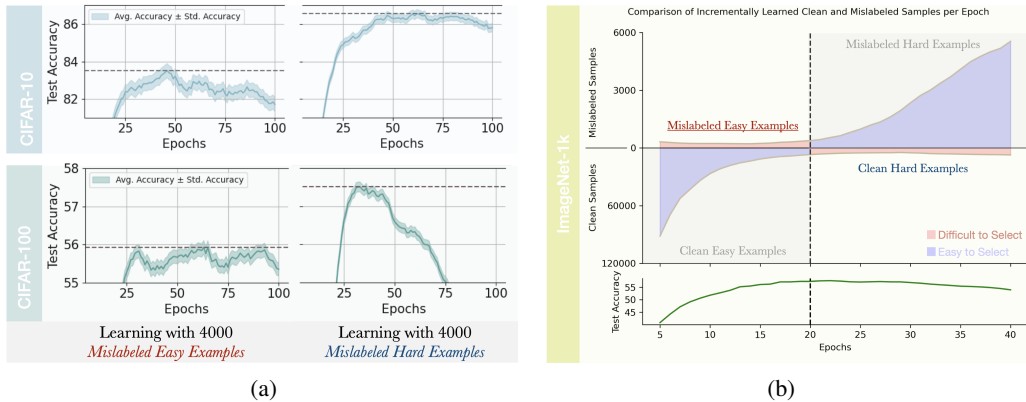

Figure 1: (a) Test accuracy curves when the originally clean training subset is augmented with 4000 *Mislabeled Easy Examples* versus 4000 *Mislabeled Hard Examples* (see Section 2.1 for setup). Adding Mislabeled Easy Examples leads to a larger decrease in the model's generalization performance. (b) Histogram illustrating the distribution of ImageNet-1k examples with 40% symmetric label noise, showing the epoch at which each example is first correctly predicted by the model during training. The horizontal axis represents the epoch when examples are first correctly predicted, and the vertical axis represents the number of examples predicted correctly at each epoch.

Although these sample selection methods have achieved decent performance by relying on early training stages to minimize noise in the selected subset and adopting advanced strategies to retain CHEs, they often overlook that not all mislabeled examples harm the model's performance equally. Specifically, even with a low noise rate in the selected subset, the presence of certain mislabeled samples can still significantly impair the model's generalization performance. As shown in Figure 1(a), we demonstrate that mislabeled samples which are correctly predicted by the model early in the training process disproportionately degrade performance. We refer to these easily learned and particularly harmful mislabeled samples as *Mislabeled Easy Examples* (MEEs). In our analysis (see Section 2.2), we find that MEEs are often closer to the centers of their mislabeled classes in the feature space of classifiers trained in the early stages. This causes them to be easily and "reasonably" classified into the wrong classes during early training, thereby disrupting the model's early learning of simple patterns (Arpit et al., 2017). Consequently, these examples are learned earlier by the model and harm generalization performance more.

To address this issue, we propose a novel sample selection strategy called *Early Cutting*, which introduces a recalibration step using the model's state at a later epoch to re-select the confident subset of samples identified during early learning. In this recalibration step, we identify samples that exhibit high loss yet are predicted with high confidence and demonstrate low sensitivity to input perturbations—characteristics indicative of MEEs. By further excluding these deceptive samples from the confident subset, we reduce MEEs negative impact on the model's generalization performance. Although this re-selection might result in the inadvertent removal of some clean samples, the impact is mitigated due to the nature of early-learned samples, which are abundant and often redundant representations of simple patterns. Removing a portion of these samples has a smaller detrimental effect compared to the significant harm caused by retaining MEEs.

We conduct extensive experiments on CIFAR (Krizhevsky et al., 2009), WebVision (Li et al., 2017), and full ImageNet-1k (Deng et al., 2009) datasets with different types and levels of label noise. The results demonstrate that our proposed method consistently outperforms state-of-the-art sample selection methods across various computer vision tasks.

Our main contributions can be summarized as follows:

1. We discover that mislabeled samples correctly predicted by the model early in training disproportionately harm model's performance; we define these samples as *Mislabeled Easy Examples* (MEEs). MEEs are closer to the centers of their mislabeled classes in the feature space of models in early training stages, causing the model to easily learn incorrect patterns.

2. We introduce *Early Cutting* method, which recalibrates the confident data subset identified early in training by utilizing the model from later stages—a counterintuitive approach, given that later-stage models are typically regarded as less trustworthy.

**Related Work.** We briefly review the related work. Detailed reviewing is in Appendix A.

*Sample Selection* has been used in learning with noisy labels to improve the robustness of model training by prioritizing confident samples. An in-depth understanding of deep learning models, particularly their learning dynamics, has facilitated research in this area. Extensive studies on the *Learning Dynamics* of DNNs have revealed that difficult clean examples are typically learned in the later stages of training (Arpit et al., 2017; Yuan et al., 2024, 2025). This insight has led to training-time metrics that quantify sample "hardness", such as learning speed (Jiang et al., 2021). These metrics inspire methods that leverage learning dynamics to select clean samples (Zhou et al., 2021; Maini et al., 2022). Various forms of *Hard Label Noise* have been studied, including asymmetric noise (Scott et al., 2013), instance-dependent noise (Xia et al., 2020b), natural noise (Wei et al., 2021b), adversarially crafted labels (Zhang et al., 2024a), open-set noise (Wei et al., 2021a), and subclass-dominant noise (Bai et al., 2023). These noise types are designed from the perspective of the labels, aiming to simulate challenging real-world scenarios or malicious attacks. In contrast to prior studies that mainly focus on different types of label noise, our work offers a fresh perspective by re-examining sample selection methods that rely on a model's early learning stages. We demonstrate that some samples hidden among those considered "confident" are, in fact, the most harmful. This contributes new insights into effectively identifying and handling mislabeled data.

## 2 Our Observations

In this section, we investigate the varying effects that different mislabeled examples have on model's generalization. In Section 2.1, we provide empirical evidence demonstrating that different mislabeled examples have varying impacts on the performance of model, with the mislabeled examples learned earlier by the model bring greater harm. In Section 2.2, we analyze the reasons why these examples are easily learned by the model and bring about greater harm.

### 2.1 Effects on Generalization from Mislabeled Examples Learned at Different Stages

Previous studies have shown that DNNs typically exhibit a specific learning pattern: they tend to learn simple and clean patterns first and gradually memorize more complex or mislabeled examples later (Arpit et al., 2017; Toneva et al., 2018). Based on this, some sample selection methods (Liu et al., 2020; Bai and Liu, 2021) trust the samples learned early by the model, treating them as high-probability clean samples. In our study, to distinguish the order in which the model learns different mislabeled examples, we refer to the definition in Yuan et al. (2023). Specifically, we consider that the model has learned a sample $(\mathbf{x}_i, \tilde{y}_i)$ at time $E_i$ if it consistently predicts the given label $\tilde{y}_i$ for both epoch $E_i - 1$ and $E_i$, regardless of whether the label is correct. Formally, we define the *learning time* $LT_i$ of a sample $(\mathbf{x}_i, \tilde{y}_i)$ as:

$$LT_i = \min\{E_i \mid \hat{y}_i^{E_i-1} = \hat{y}_i^{E_i} = \tilde{y}_i\}, \tag{1}$$

where $\hat{y}_i^t$ denotes the model's predicted label for instance $\mathbf{x}_i$ at epoch $e$. By tracking each sample's learning time $LT_i$, we can analyze the order in which the model learns different samples and evaluate their impact on performance.

(a)Impact on model performance from mislabeled examples learned at different stages, using CIFAR-10.

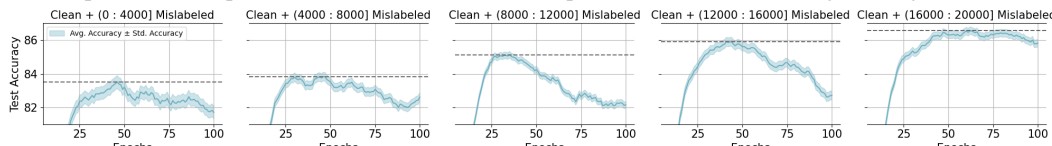

Figure 2: Impact of mislabeled samples learned at different stages on model generalization performance. Subfigure 2 shows the scenario in the CIFAR-10 dataset, which contains 20,000 mislabeled samples (40% instance-dependent label noise) and 30,000 clean samples. We divided the 20,000 mislabeled samples into five groups based on the order in which an initial model learned them—from earliest to latest (ranging from $(0 : 20,000]$). Each group was combined with the 30,000 clean samples, creating datasets with approximately 12% label noise ($4,000/34,000$). New models were then trained on these datasets. As shown by the decreasing test accuracy, models trained on datasets containing earlier-learned mislabeled samples (e.g., "Clean $+(0 : 4000]$ Mislabeled") exhibited lower generalization performance. Subfigure 6 shows similar findings on CIFAR-100.

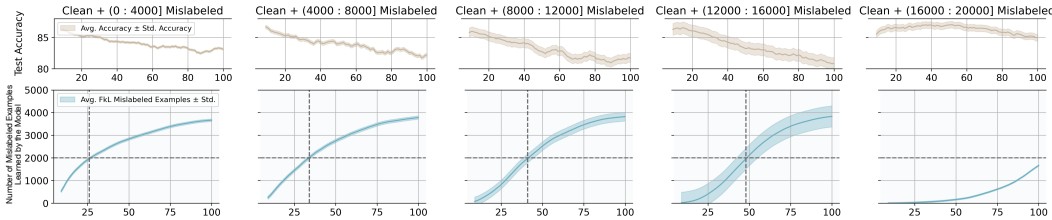

(a)The speed at which pretrained models on CIFAR-10 learn mislabeled examples from different stages.

Figure 3: Comparison of how pretrained models learn mislabeled examples from different learning stages. Subfigure 3 shows results on CIFAR-10 with 40% noise. We divided the mislabeled examples into five groups based on the order the initial model learned them, mixing each group with 30,000 clean examples to form datasets with approximately $12\%$ label noise ($4000/34000$). A model was pretrained on the 30,000 clean examples and then trained on these new noisy datasets. Reference lines indicate the number of epochs required for the pretrained model to learn different sets of 2,000 mislabeled examples. The results reveal that earlier-learned mislabeled examples are also learned more quickly by the robust model. Subfigure 7 shows similar findings on CIFAR-100.

To investigate how the learning order of mislabeled examples affects generalization, we conducted experiments on CIFAR-10 and CIFAR-100 with 40% instance-dependent label noise. First, we trained an initial model on the noisy dataset to record the learning time $LT_i$ for each sample. Based on these times, we partitioned the 20,000 mislabeled examples into five sequential groups of 4,000, from earliest-learned to latest-learned. Each group was then combined with 30,000 clean examples to form five distinct training datasets.

We then trained new models from scratch on these datasets. As shown in Figure 2, the results are unambiguous: models trained with the earliest-learned group of mislabeled examples exhibit significantly lower generalization performance than those trained with later-learned groups. This clearly indicates that mislabeled examples learned earlier by a model cause greater harm to its generalization. To validate this observation, we repeated the experiment using a model pretrained on only the clean data. As shown in Figure 3, the model still learns the MEEs from the "earliest" group much faster than those from later groups. This confirms that these samples are inherently easy for the model to learn, regardless of the training starting point.

## 2.2 Mislabeled Easy Examples

In this subsection, we focus specifically on the mislabeled examples that the model learns during the early stages of training. Drawing inspiration from the concept of *Clean Hard Examples*, we formally define these particularly harmful mislabeled examples learned early by the model as *Mislabeled Easy Examples (MEEs)*. This term indicates that although these samples are incorrectly labeled, they are easily learned by the model.

Notably, MEEs are non-trivial because the early stages of model training are typically characterized by learning simple and correct patterns from clean samples (Arpit et al., 2017; Toneva et al., 2018), while the later stages are when the model starts to memorize mislabeled samples (Zhang et al., 2021; Yuan et al., 2024). Therefore, it is worthwhile to conduct an in-depth exploration of the counterintuitive way in which the model learns these mislabeled samples early in training to enhance our understanding of its learning process. To better understand the characteristics of MEEs and their impact on model generalization, we examine their positions in the model's feature space (the representation before the last fully-connected layer) and present some representative examples.

As shown in the Figure 4(a), we visualize the mislabeled examples that are correctly predicted by the early-stage model using t-SNE (Van der Maaten and Hinton, 2008) in the feature space. For a detailed visualization of how this feature space and the associated distance ratios evolve at later training stages, please see Appendix E. Further, to quantify the model's representations of mislabeled samples during early training, we compute the Euclidean distances from each mislabeled example learned by the early-stage model to the center of its *true* class and the center of its *mislabeled* class in the embedded feature space. We denote these two distances as $d_{\text{true}}$ and $d_{\text{mislabeled}}$, respectively. We then define the *distance ratio* $r = d_{\text{mislabeled}}/d_{\text{true}}$. If $r < 1$, the example is closer to the *mislabeled* class center than to its *true* class center. As shown in the bottom row, MEEs exhibit a notably smaller

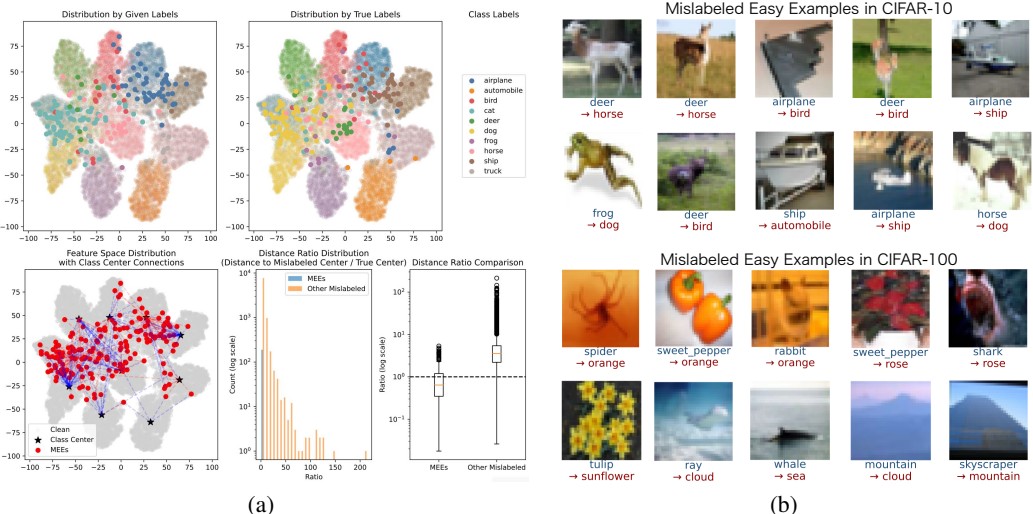

Figure 4: 4(a) Visualization of *Mislabeled Easy Examples* (MEEs) in the feature space. Top row: t-SNE embeddings of CIFAR-10 training samples (20% instance-dependent label noise), colored by their *given* labels (left) and their *true* labels (middle). Bottom left: a closer look at MEEs (red points) connected to their mislabeled class centers (black stars), demonstrating how these examples cluster in ambiguous regions that overlap with the mislabeled class. Bottom middle and right: comparisons of the distance ratio $r = d_{\text{mislabeled}}/d_{\text{true}}$ for MEEs and other mislabeled samples, confirming that they are indeed closer to incorrect wrong labels than their true labels in the learned feature space. 4(b) Representative MEEs. Each image is shown with its *true* label (blue) and the *mislabeled* label (red).

median distance ratio (0.830), with more than half (53.8%) of them having $r < 1$. In contrast, the remaining mislabeled samples (non-MEEs) have a median ratio of 3.923, and only 5.4% are closer to the incorrect class.

**Why MEEs are learned earlier and harm generalization more?** Our analysis suggests that MEEs occupy regions in the feature space where their incorrect labels seem more reasonable to the model. During the early stages of training, instances of MEEs closely resemble their mislabeled classes in the feature space, the model learned them as if they were representative samples with simple patterns. Figure 4(b) presents representative MEEs. For instance, a CIFAR-10 image of an airplane with a dominant sea background is mislabeled as a *ship*, and a CIFAR-100 image featuring a predominantly orange background is mislabeled as an *orange*. These examples illustrate how strong visual cues matching their given (incorrect) label classes—such as color, texture, or prominent features—can pull these samples closer to the incorrect class in the feature space.

This phenomenon explains why MEEs are learned earlier: their misleading features align with the simple patterns of their given (incorrect) label classes that the model is tend to learn during the initial training stages. Thus, the early learning of MEEs has a disproportionately negative impact on the model's generalization performance: since the model incorporates incorrect patterns associated with MEEs from the beginning, it disrupts the initial formation of simple and accurate feature representations. The erroneous features learned from MEEs become intertwined with the representations of clean data, making it challenging for the model to disentangle the clean patterns. To further quantify this detrimental impact at a microscopic level, we conducted a supplementary analysis using influence functions (Koh and Liang, 2017) to measure the precise impact of individual samples on model generalization. This analysis, detailed in Appendix B, provides direct quantitative evidence that MEEs are substantially more harmful to the model's performance on clean data than other mislabeled examples.

## 3 Methodology

Based on the analysis above, MEEs, mislabeled samples that the model learns easily during early training stages, can have a disproportionately negative impact on model generalization performance. Previous methods (Liu et al., 2020; Bai and Liu, 2021) often rely on trusting the model's early learning

stages or focusing on samples with small loss, are ineffective at filtering out MEEs due to their deceptive nature. To mitigate the influence of MEEs, we propose a novel sample selection strategy called *Early Cutting*. Early Cutting leverages the model's state at a later training phase—specifically, at the early stopping epoch $t$ when the model begins to overfit—to re-evaluate and refine the subset of samples initially identified as confidently learned. The initial confident subset, $\mathcal{D}^s$, is formed by selecting samples with small *learning times* $LT_i$ (as defined in Eq. (1)). This captures examples that are learned quickly, encompassing both genuinely clean, easy samples and, crucially, the MEEs.

From this initially selected confident subset $\mathcal{D}^s$, our objective is to identify and subsequently remove samples that we characterize as MEEs in Section 2.2. These are samples that, when evaluated by the model $f_{\theta^t}$ at the early stopping epoch $t$, exhibit predictions that differ from their given noisy labels $\tilde{y}_i$), yet are made with high confidence and possess stable gradients.

Formally, consider $\mathcal{D}^s = (\mathbf{x}_i, \tilde{y}_i)_{i=1}^N$, where $\mathbf{x}_i \in \mathbb{R}^d$ represents input features, and $\tilde{y}_i \in \mathcal{Y}$ denotes the corresponding observed labels from $K$ classes. Let $f_{\theta^t}(\mathbf{x})$ be a model parameterized by $\theta^t$ at the early stopping epoch $t$, generating class probabilities via the softmax function:

$$\mathbf{p}_i = f_{\theta^t}(\mathbf{x}_i) = \left[ p_i^{(1)},\ p_i^{(2)},\ \ldots,\ p_i^{(K)} \right], \tag{2}$$

where $p_i^{(k)}$ is the model's output probability for class k. The predicted label $\hat{y}_i$ and the prediction confidence $c_i$ are given by: $\hat{y}_i = \arg\max_k p_i^{(k)}$, and $c_i = p_i^{(\hat{y}_i)}$. The cross-entropy loss for sample $(\mathbf{x}_i, \tilde{y}_i)$ is: $L_i = -\log p_i^{(\tilde{y}_i)}$.

While the selected subset $\mathcal{D}^s$ tends to retain a high-quality set of clean samples, it may still include MEEs due to their deceptive nature. To address this issue, we leverage the model's parameters $\theta^t$ at a later training stage $t$ to identify and remove suspicious samples from $\mathcal{D}^s$. Specifically, we define a set of suspicious samples $\mathcal{S}$ within $\mathcal{D}^s$ based on the criteria of high loss and high confidence:

$$\mathcal{S} = \left\{ i \in \mathcal{D}^s \mid L_i > \delta, \quad c_i > \tau \right\}, \tag{3}$$

where $\delta$ and $\tau$ are thresholds for the loss and confidence, respectively. The rationale is that a high loss $L_i$ indicates that the model's prediction at epoch $t$ disagrees with the given label $\tilde{y}_i$, and a high confidence $c_i$ implies that the model is very certain about its (contradictory) prediction $\hat{y}_i \neq \tilde{y}_i$. Therefore, samples satisfying both conditions are likely to be mislabeled, even if they were learned early. Our method remains robust even if the set $\mathcal{S}$ is empty, as the refinement step would simply be bypassed. Relying solely on loss and confidence may not be sufficient, as some hard-to-learn samples may also exhibit high loss and high confidence due to their intrinsic difficulty. To further refine our selection, we introduce the concept of gradient stability. We compute the Euclidean norm of the gradient of the loss $L_i$ with respect to the input $\mathbf{x}_i$:

$$\nabla_{\mathbf{x}_i} L_i = \frac{\partial L_i}{\partial \mathbf{x}_i}, \quad g_i = \|\nabla_{\mathbf{x}_i} L_i\|_2. \tag{4}$$

A small gradient norm $g_i$ indicates that the loss $L_i$ is insensitive to small perturbations in $\mathbf{x}_i$, suggesting a strong (but potentially incorrect) association between the input features and the predicted label. MEEs tend to have low gradient norms because the model has confidently mislearned them, making the loss stable even under input perturbations. We refine $\mathcal{S}$ by selecting samples with high gradient stability (where $\epsilon$ is a threshold):

$$\mathcal{S}' = \left\{ i \in \mathcal{S} \mid g_i < \epsilon \right\}. \tag{5}$$

**Operational Definition 1** (Mislabeled Easy Examples (MEEs)). *Operationally, from the set of early-learned samples $\mathcal{D}^s$, we define the subset of Mislabeled Easy Examples (MEEs) as those samples $i$ that satisfy the conditions of high loss, high confidence, and low input gradient norm. Formally:*

$$\textit{MEEs} = \{ i \in \mathcal{D}^s \mid (L_i > \delta) \wedge (c_i > \tau) \wedge (g_i < \epsilon) \}. \tag{6}$$

This identification is practically implemented by selecting samples whose metrics fall into predefined percentiles derived from their distributions within $\mathcal{D}^s$. Specifically, for all settings, we target the top 10% for loss, top 20% for confidence, and bottom 20% for gradient norm, which were determined on a validation set. Additionally, we set the early cutting rate to 1.5. Samples meeting these criteria are classified as MEEs and subsequently removed from $\mathcal{D}^s$, yielding a refined subset for further training. We formalize the complete *Early Cutting* procedure in Algorithm 1.

---

**Algorithm 1** Iterative Sample Selection with Early Cutting

---

**Require:** Training data $\mathcal{D}^0$; Number of iterations $I_{\text{rate}}$; Early cutting rate $\gamma$; Thresholds $\delta, \tau, \epsilon$
**Ensure:** Trained model parameters $\theta^*$
1: **for** $i = 1$ to $I_{\text{rate}}$ **do**
2:     *1. Base Sample Selection:*
3:     Compute learning times $LT_j$ for all $(\mathbf{x}_j, \tilde{y}_j) \in \mathcal{D}^{i-1}$ using Eq. (1)
4:     Select the initial early-learned subset $\mathcal{D}^s$ based on smallest learning times.
5:     *2. Early Cutting:*
6:     Select candidate subset $\mathcal{D}'^s$ from $\mathcal{D}^s$ (e.g., the $\frac{1}{\gamma}$ proportion with the earliest learning times).
7:     Compute loss $L_j$, confidence $c_j$, and gradient norm $g_j$ for all samples in $\mathcal{D}'^s$.
8:     Identify the set of MEEs in $\mathcal{D}'^s$ according to Definition 1:
9:         MEEs $= \left\{ j \in \mathcal{D}'^s \mid (L_j > \delta) \wedge (c_j > \tau) \wedge (g_j < \epsilon) \right\}$
10:     Create the refined subset by removing the identified MEEs: $\mathcal{D}^s_{\text{refined}} \leftarrow \mathcal{D}^s \setminus \text{MEEs}$.
11:     Update the training data for the next iteration: $\mathcal{D}^i \leftarrow \mathcal{D}^s_{\text{refined}}$.
12: **end for**
13: **Final Training Phase:** Train a model from scratch on the final refined set $\mathcal{D}^{I_{\text{rate}}}$ until convergence.
14: **return** Trained model parameters $\theta^*$.

---

Notably, the proposed method operates on $\mathcal{D}^s$, the subset of samples learned quickly during the initial training phase. Such early-learned subsets are known to often contain significant redundancy, with multiple examples representing similar, dominant data patterns (Feldman, 2020; Feldman and Zhang, 2020; Yuan et al., 2024). This inherent redundancy contributes to the robustness of our MEEs removal strategy. Firstly, it provides resilience against the inadvertent removal of a small number of clean samples, as their informational content is likely preserved by other remaining examples. Secondly, this characteristic makes the outcome less sensitive to the precise percentile thresholds used for MEEs selection. A sensitivity analysis of these thresholds is presented in Section 4.3, an ablation study is shown in Appendix D.8, and it transferability is shown in Appendix D.9.

## 4 Experiments

### 4.1 Preliminary Presentation of Effectiveness

We first provide empirical evidence to verify the effectiveness of *Early Cutting*. Using CIFAR-10 with 40% various synthetic label noise and ResNet-18 as the backbone, we compared our proposed *Early Cutting* with *loss-based* and *dynamic-based* sample selection methods. Table 1 shows that *Early Cutting* consistently achieved the highest test accuracy across all noise types. Although *Early Cutting* and the *dynamic-based* method selected training subsets with similar noise rates, our approach's better performance indicates that focusing on filtering specific harmful mislabeled examples improves selection quality. The last row shows the number of additional samples filtered by *Early Cutting* and the high percentage of mislabeled samples among them, proving its effectiveness at identifying and removing noisy labels. As intuition, more challenging noise types result in more mislabeled samples being removed, leading to larger performance gains. Detailed settings in Appendix D.

Table 1: Training on 60% noisy training samples selected by each method. Test accuracy (*noise rates in selected training samples*).

|  | Symmetric 40% | Asymmetric 40% | Pairflip 40% | Instance. 40% |
|---|---|---|---|---|
| *Loss-based* Selection (Han et al., 2018) | 83.01% (*10.44%*) | 83.79% (*4.84%*) | 84.16% (*10.88%*) | 82.87% (*11.11%*) |
| *Dynamic-based* Selection (Yuan et al., 2023) | 89.39% (*4.57%*) | 84.28% (*3.37%*) | 84.71% (*10.19%*) | 83.12% (*12.52%*) |
| *Dynamic-based + Early Cutting* Selection | **89.66%** (***4.94%***) | **84.85%** (***3.33%***) | **85.88%** (***9.52%***) | **84.31%** (***12.06%***) |
| Additional Samples Filtered by *Early Cutting* | 98 (*56.12%*) | 191 (*95.29%*) | 161 (*45.96%*) | 300 (*91.33%*) |

### 4.2 Comparison with the Competitors

**Competitors.** We compare our approach with several methods: robust loss functions including *GCE* (Zhang and Sabuncu, 2018) and *Student Loss* (Zhang et al., 2024b); robust training methods, including *Co-teaching* (Han et al., 2018) and *CSGN* (Lin et al., 2024b); and sample selection methods,

Table 2: Test performance (mean±std) of each approach using ResNet-18 on CIFAR-10.

| | Symmetric 20% | Symmetric 40% | Instance. 20% | Instance. 40% |
|---|---|---|---|---|
| Cross-Entropy | $86.64 \pm 0.18\%$ | $82.64 \pm 0.29\%$ | $87.62 \pm 0.09\%$ | $82.82 \pm 0.37\%$ |
| GCE (Zhang and Sabuncu, 2018) | $91.50 \pm 0.33\%$ | $87.02 \pm 0.16\%$ | $89.42 \pm 0.31\%$ | $83.10 \pm 0.29\%$ |
| Co-teaching (Han et al., 2018) | $89.13 \pm 0.38\%$ | $82.29 \pm 0.21\%$ | $89.42 \pm 0.22\%$ | $81.91 \pm 0.20\%$ |
| Me-Momentum (Bai and Liu, 2021) | $92.76 \pm 0.15\%$ | $90.75 \pm 0.49\%$ | $91.87 \pm 0.22\%$ | $88.80 \pm 0.29\%$ |
| Self-Filtering (Wei et al., 2022) | $92.88 \pm 0.22\%$ | $90.46 \pm 0.28\%$ | $92.35 \pm 0.13\%$ | $86.93 \pm 0.14\%$ |
| VOG (Agarwal et al., 2022) | $87.90 \pm 0.22\%$ | $84.37 \pm 0.21\%$ | $87.71 \pm 0.15\%$ | $82.52 \pm 0.13\%$ |
| Late Stopping (Yuan et al., 2023) | $92.02 \pm 0.17\%$ | $88.25 \pm 1.01\%$ | $91.65 \pm 0.26\%$ | $88.28 \pm 0.24\%$ |
| Misdetect (Deng et al., 2024) | $92.20 \pm 0.38\%$ | $87.31 \pm 0.30\%$ | $88.44 \pm 0.51\%$ | $85.11 \pm 0.42\%$ |
| RLM (Li et al., 2024) | $93.11 \pm 0.29\%$ | $91.06 \pm 0.17\%$ | $93.13 \pm 0.05\%$ | $89.73 \pm 0.32\%$ |
| Student Loss (Zhang et al., 2024b) | $91.90 \pm 0.37\%$ | $89.03 \pm 0.32\%$ | $89.99 \pm 0.50\%$ | $81.95 \pm 0.51\%$ |
| CSGN (Lin et al., 2024b) | $90.09 \pm 0.32\%$ | $87.71 \pm 0.46\%$ | $89.45 \pm 0.07\%$ | $88.50 \pm 0.49\%$ |
| Early Cutting (Ours) | $\mathbf{93.79 \pm 0.14\%}$ | $\mathbf{91.80 \pm 0.18\%}$ | $\mathbf{93.40 \pm 0.22\%}$ | $\mathbf{90.78 \pm 0.31\%}$ |

Table 3: Test performance (mean±std) of each approach using ResNet-34 on CIFAR-100.

| | Symmetric 20% | Symmetric 40% | Instance. 20% | Instance. 40% |
|---|---|---|---|---|
| Cross-Entropy | $63.04 \pm 0.41\%$ | $51.81 \pm 0.33\%$ | $63.36 \pm 0.22\%$ | $51.58 \pm 0.96\%$ |
| GCE (Zhang and Sabuncu, 2018) | $66.68 \pm 0.35\%$ | $59.42 \pm 0.19\%$ | $64.71 \pm 0.15\%$ | $55.49 \pm 0.34\%$ |
| Co-teaching (Han et al., 2018) | $66.72 \pm 0.26\%$ | $58.72 \pm 0.43\%$ | $66.45 \pm 0.28\%$ | $59.52 \pm 0.32\%$ |
| Me-Momentum (Bai and Liu, 2021) | $71.94 \pm 0.27\%$ | $67.36 \pm 0.30\%$ | $72.47 \pm 0.39\%$ | $63.99 \pm 0.56\%$ |
| Self-Filtering (Wei et al., 2022) | $70.18 \pm 0.39\%$ | $66.92 \pm 0.18\%$ | $69.52 \pm 0.38\%$ | $66.76 \pm 0.42\%$ |
| VOG (Agarwal et al., 2022) | $66.78 \pm 0.21\%$ | $60.55 \pm 0.40\%$ | $66.81 \pm 0.23\%$ | $56.57 \pm 0.32\%$ |
| Late Stopping (Yuan et al., 2023) | $71.09 \pm 0.71\%$ | $65.43 \pm 0.50\%$ | $70.32 \pm 0.06\%$ | $61.71 \pm 0.25\%$ |
| Misdetect (Deng et al., 2024) | $73.90 \pm 0.34\%$ | $65.10 \pm 0.40\%$ | $70.45 \pm 0.14\%$ | $63.66 \pm 0.17\%$ |
| RLM (Li et al., 2024) | $71.68 \pm 0.32\%$ | $67.68 \pm 0.36\%$ | $68.26 \pm 0.37\%$ | $67.31 \pm 0.64\%$ |
| Student Loss (Zhang et al., 2024b) | $69.04 \pm 0.19\%$ | $64.21 \pm 0.49\%$ | $67.62 \pm 0.67\%$ | $56.24 \pm 0.24\%$ |
| CSGN (Lin et al., 2024b) | $69.89 \pm 0.22\%$ | $56.18 \pm 0.36\%$ | $71.97 \pm 0.10\%$ | $65.43 \pm 0.52\%$ |
| Early Cutting (Ours) | $\mathbf{76.20 \pm 0.27\%}$ | $\mathbf{72.77 \pm 0.17\%}$ | $\mathbf{75.03 \pm 0.23\%}$ | $\mathbf{69.94 \pm 0.30\%}$ |

Table 4: Test performance (mean±std) of each approach using ResNet-18 and 34 on CIFAR-N.

| | 10N Random 1 | 10N Random 2 | 10N Random 3 | 10N Worst | 100N Fine |
|---|---|---|---|---|---|
| Cross-Entropy | $86.16 \pm 0.14\%$ | $85.74 \pm 0.28\%$ | $85.91 \pm 0.14\%$ | $80.00 \pm 0.42\%$ | $54.53 \pm 0.13\%$ |
| Late Stopping (Yuan et al., 2023) | $89.71 \pm 0.73\%$ | $90.23 \pm 0.37\%$ | $90.49 \pm 0.31\%$ | $86.10 \pm 0.41\%$ | $57.32 \pm 0.19\%$ |
| RLM (Li et al., 2024) | $92.21 \pm 0.37\%$ | $92.27 \pm 0.31\%$ | $92.07 \pm 0.72\%$ | $86.25 \pm 0.24\%$ | $57.90 \pm 0.33\%$ |
| Student Loss (Zhang et al., 2024b) | $90.60 \pm 0.07\%$ | $90.44 \pm 0.28\%$ | $90.44 \pm 0.35\%$ | $86.16 \pm 0.31\%$ | $58.55 \pm 0.53\%$ |
| CSGN (Lin et al., 2024b) | $89.14 \pm 0.23\%$ | $89.49 \pm 0.25\%$ | $89.25 \pm 0.31\%$ | $82.88 \pm 0.51\%$ | $58.13 \pm 0.49\%$ |
| Early Cutting (Ours) | $\mathbf{92.50 \pm 0.14\%}$ | $\mathbf{92.65 \pm 0.11\%}$ | $\mathbf{92.36 \pm 0.43\%}$ | $\mathbf{87.43 \pm 0.13\%}$ | $\mathbf{66.52 \pm 0.22\%}$ |

Table 5: Test performance of each approach using ResNet-50 on large-scale naturalistic datasets.

| | WebVision Validation | ILSVRC12 Validation | Full ImageNet-1k (Sym. 40%) |
|---|---|---|---|
| Cross-Entropy | 67.32% | 63.84% | 67.99% |
| Late Stopping (Yuan et al., 2023) | 71.56% | 68.32% | 71.42% |
| RLM (Li et al., 2024) | 72.28% | 69.86% | 68.95% |
| Student Loss (Zhang et al., 2024b) | 69.80% | 67.62% | 69.44% |
| CSGN (Lin et al., 2024b) | 72.32% | 69.52% | - |
| Early Cutting (Ours) | **73.81%** | **71.20%** | **73.28%** |

including *Me-Momentum* (Bai and Liu, 2021), *Self-Filtering* (Wei et al., 2022), *VOG* (Agarwal et al., 2022), *Late Stopping* (Yuan et al., 2023), *Misdetect* (Deng et al., 2024) and *RLM* (Li et al., 2024).

**Datasets and implementation.** We conducted experiments on several benchmark datasets to evaluate our proposed method compare with above competitors. For synthetic noise experiments, we used *CIFAR-10* and *CIFAR-100* (Krizhevsky et al., 2009), adding *symmetric* and *instance-dependent* label noise at rates of 20% and 40% following standard protocols (Bai and Liu, 2021; Yuan et al., 2023). We split 10% noisy trianing data for validation. For real-world noisy labels, we utilized *CIFAR-N* (Wei et al., 2021b), as well as the large-scale *WebVision* dataset (Li et al., 2017). Following previous work (Lin et al., 2024b; Li et al., 2024), we used the first 50 classes of the *WebVision* dataset and validated on both the *WebVision* validation set and the *ILSVRC12* (Russakovsky et al., 2015) validation set. We further confirmed the scalability of our proposed method on the full *ImageNet-1K* (Deng et al., 2009) with 40% synthetic symmetric label noise. Training was performed using *SGD* (Robbins and Monro, 1951) with a momentum (Rumelhart et al., 1986) of 0.9 and a weight decay (Krogh and Hertz, 1991) of $5 \times 10^{-4}$. The initial learning rate was set to 0.1 and decayed using a cosine annealing schedule.

Table 6: Test accuracy comparison of different approaches using semi-supervised learning.

| Methods | SSL | CIFAR-10 | | CIFAR-100 | |
|---|---|---|---|---|---|
| | | Symmetric 50% | Instance. 40% | Symmetric 50% | Instance. 40% |
| Early Cutting (Ours) | - | 90.3% | 90.7% | 69.6% | 69.9% |
| CORES$^{2*}$ (Cheng et al., 2020) | UDA | 93.1% | 92.2% | 73.1% | 71.9% |
| Divide-Mix (Li et al., 2020) | MixMatch | 94.6% | 93.0% | 74.6% | 71.7% |
| ELR+ (Liu et al., 2020) | MixMatch | 93.8% | 92.2% | 72.4% | 72.6% |
| SFT+ (Wei et al., 2022) | MixMatch | 94.9% | 94.1% | 75.2% | 74.6% |
| RLM+ (Li et al., 2024) | MixMatch | 95.1% | 94.8% | 72.9% | 72.8% |
| Early Cutting+ (Ours) | MixMatch | **95.8%** | **95.5%** | **75.6%** | **75.4%** |

Models were trained (for the final iteration) for 300 epochs on the *CIFAR* datasets, for 200 epochs on *WebVision*, and for 150 epochs on full *ImageNet-1k*. We re-implemented all competitor methods with consistent settings (unless otherwise specified). For all experiments reporting mean±std, the results are the average and standard deviation of three trials using different random seeds. Detailed settings are provided in Appendix D.

**Discussions on experimental results.** As shown in Tables 2, 3, 4, and 5, our proposed *Early Cutting* method consistently achieves outstanding performance across various datasets and noise conditions. On standard benchmarks like *CIFAR*, it attains the highest test accuracy regardless of the type of label noise—symmetric, instance-dependent, or real-world—and across different noise rates. Notably, it performs exceptionally well on *CIFAR-100*, which has a larger number of classes, indicating strong robustness in handling label noise in fine-grained classification tasks. Without further fine-tuning of hyperparameters, *Early Cutting* also achieves significant performance improvements on large-scale datasets such as *WebVision* and full *ImageNet-1k*. This demonstrates the practicality and scalability of our method in handling challenging scenarios. By iteratively selecting confident samples and removing harmful mislabeled easy examples, our method helps the model learn from reliable data while avoiding overconfidence in early-learned samples. Furthermore, to demonstrate the transferability of our method to different model architectures, we conducted additional experiments on the transformer-based TinyViT (Vaswani et al., 2017; Wu et al., 2022), which are detailed in Appendix D.9. Appendix D.7 provides a training time evaluation, demonstrating that our proposed Early Cutting method, executed once per training round, incurs an additional computational overhead of less than one percent of the total training duration. This is significantly lower than the base computational cost associated with selecting the initial confident subset $\mathcal{D}^s$ in each training round.

## 4.3 Further Analysis

**Semi-supervised learning.** To further evaluate the effectiveness of our *Early Cutting* method, we integrated it with the *MixMatch* (Berthelot et al., 2019) semi-supervised learning framework, resulting in *Early Cutting+*. We treat the confident samples obtained from sample selection as labeled data, and the samples removed from training in the fully supervised setting as unlabeled data. The confident set (labeled data) and the non-confident set (unlabeled data) are identified once and remain fixed throughout the subsequent *MixMatch* training. We compared *Early Cutting+* with advanced SSL-based LNL methods, including CORES$^{2*}$ (Cheng et al., 2020), DivideMix (Li et al., 2020), and ELR+ (Liu et al., 2020); additionally, we compared with the latest sample selection methods integrating SSL, SFT+ (Wei et al., 2022) and RLM+ (Li et al., 2024). Some baseline results are taken from Wei et al. (2022). As shown in Table 6, our *Early Cutting+* achieves the highest test accuracy on both *CIFAR-10* and *CIFAR-100* under 50% symmetric and 40% instance-dependent label noise, surpassing previous methods. These results underscore the capability of *Early Cutting* in selecting high-quality subsets of training samples.

**Sensitivity analysis.** We conducted sensitivity analyses to evaluate the robustness of our *Early Cutting* method. As shown in Figure 5, our method achieves optimal test accuracy on both *CIFAR-10* and *CIFAR-100* with 40% symmetric label noise when using the same default thresholds (also used for *WebVision* and full *ImageNet-1k*) for identifying samples with large loss, high confidence, low gradient norm, and early cutting rate. Notably, even when the hyperparameters vary over a wide range, our method exhibits minimal sensitivity, with only slight ($< 1\%$) performance degradation. Results indicate that our method is robust and effective across different datasets without requiring extensive hyperparameter tuning. A further ablation study is presented in Appendix D.8.

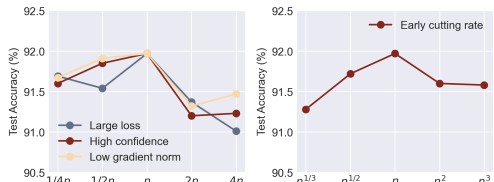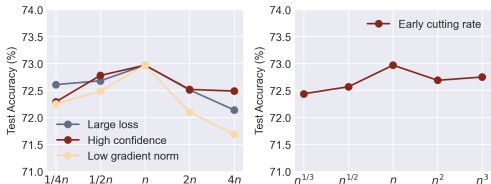

| (a) CIFAR-10 with 40% symmetric label noise | (b) CIFAR-100 with 40% symmetric label noise |

Figure 5: Sensitivity analysis of hyperparameters on CIFAR-10 and CIFAR-100 with 40% symmetric label noise. In each subfigure, the left plot shows test accuracy versus thresholds for the large loss, high confidence, and low gradient norm criteria, scaled by factors of $\frac{1}{4}$, $\frac{1}{2}$, 1, 2, and 4. The right plot shows test accuracy versus Early Cutting rate $\gamma$ set to $n^{1/3}$, $n^{1/2}$, $n$, $n^2$, and $n^3$, where $\gamma \geq 1$.

## 5 Conclusion

In this paper, we uncovered an oversight in existing methods for learning with noisy labels by demonstrating that not all mislabeled examples harm the model's performance equally. We identified a specific subset termed *Mislabeled Easy Examples* (MEEs)—mislabeled samples that the model learns early and that significantly mislead the training process. To address this issue, we proposed *Early Cutting*, a counter-intuitive sample selection strategy that recalibrates the confident subset by leveraging the model's later training state, which is typically considered unreliable, to effectively filter out MEEs. This work provides a practical solution for learning with noisy labels and advances the understanding of how different mislabeled samples affect deep learning models.

**Limitation.** We acknowledge several limitations that suggest avenues for future work. First, our empirical validation is confined to visual classification tasks, and the method's applicability to other domains such as natural language processing or tabular data remains to be explored. Second, while our work provides strong empirical evidence for the detrimental impact of MEEs, this justification is primarily observational; a formal theoretical analysis explaining why these specific examples disproportionately harm generalization would further strengthen our claims. Third, as an iterative strategy, our method introduces computational overhead by requiring re-evaluation of samples and computation of metrics like input gradients at later training stages (see Appendix D.7 for a detailed analysis). Finally, potential failure modes may arise in extreme data scenarios. In datasets with severe label noise or significant class imbalance (e.g., long-tailed distributions), the distinction between MEEs and clean hard examples from rare classes could become ambiguous. This poses a risk of inadvertently filtering out valuable samples from minority classes, which could potentially introduce fairness concerns by biasing the final model against underrepresented groups.

## Acknowledgments

The authors would like to thank the anonymous reviewers for their insightful and constructive comments. The authors are also grateful to the Area Chairs for their diligent work. During the preparation of this paper, the authors utilized large language models to enhance the clarity of the writing and to generate code for training and visualization. TL is partially supported by the following Australian Research Council projects: FT220100318, DP220102121, LP220100527, LP220200949. BH was supported by RGC Young Collaborative Research Grant No. C2005-24Y and RGC General Research Fund No. 12200725.

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

# A  Related Work

**Learning with Noisy Labels (LNL)** has been an active research area in recent years (Wang et al., 2022; Gong et al., 2023; Baek et al., 2024; Englesson and Azizpour, 2024; Huang et al., 2024), with numerous methods proposed to mitigate the impact of label noise on deep neural networks (DNNs). Formally, let $X$ denote the input space, and let $\mathcal{Y} = \{1, 2, \ldots, K\}$ be the set of possible labels. Let $Y$ be the random variable for the clean label and $\tilde{Y}$ be the random variable for the observed noisy label, both taking values in $\mathcal{Y}$. Consider the clean data distribution $P(X, Y)$, from which clean samples $(\mathbf{x}, y)$ are drawn. In practice, we often have access only to a training dataset with potentially noisy labels:

$$\tilde{D} = \{(\mathbf{x}_i, \tilde{y}_i)\}_{i=1}^{n}, \tag{7}$$

where $\mathbf{x}_i \in X$ and $\tilde{y}_i \in \mathcal{Y}$ are observed noisy labels. The aim is to learn a robust classifier $f : X \to \mathcal{Y}$ parameterized by $\theta$, which performs well on clean test data drawn from the distribution $P(X, Y)$. The noise process is typically modeled using a noise transition matrix $T \in \mathbb{R}^{K \times K}$, defined as:

$$T_{ij} = P(\tilde{Y} = j \mid Y = i), \quad \text{for } i, j \in \mathcal{Y}, \tag{8}$$

which represents the probability that a clean label $y = i$ is flipped to a noisy label $\tilde{y} = j$. The relationship between the clean and noisy label distributions can be expressed as:

$$P(\tilde{Y} = \tilde{y} \mid X = \mathbf{x}) = \sum_{k \in \mathcal{Y}} T_{k\tilde{y}} \, P(Y = k \mid X = \mathbf{x}). \tag{9}$$

In the context of deep learning, the classifier $f_\theta(\mathbf{x})$ is often trained by minimizing the empirical risk over the noisy dataset:

$$\min_\theta \frac{1}{n} \sum_{i=1}^{n} \ell\left(f_\theta(\mathbf{x}_i), \tilde{y}_i\right), \tag{10}$$

where $\ell(\cdot, \cdot)$ is a loss function, such as the cross-entropy loss:

$$\ell\left(f_\theta(\mathbf{x}), \tilde{y}\right) = -\log\left(f_\theta^{(\tilde{y})}(\mathbf{x})\right), \tag{11}$$

and $f_\theta^{(\tilde{y})}(\mathbf{x})$ denotes the predicted probability for class $\tilde{y}$. However, due to label noise, directly minimizing this loss can lead to the model overfitting to noisy labels, degrading its performance on clean data. To address this issue, various strategies have been proposed. In the following discussion, we focus on heuristic methods, specifically sample selection techniques, which do not rely on the explicit estimation of $T$ but instead incorporate strategies to mitigate the impact of noisy labels.

**Sample selection strategies.** Sample selection has been widely used in learning with noisy labels to improve the robustness of model training by prioritizing confident samples. An in-depth understanding of deep learning models, particularly their learning dynamics, has facilitated research in this area. Extensive studies on the *learning dynamics* of DNNs have revealed that difficult clean examples are typically learned in the later stages of training (Arpit et al., 2017; Toneva et al., 2018; Lin et al., 2024a).

In general, sample selection methods assign a statistical characteristic to each sample and select a subset of samples that fall below a certain threshold (Han et al., 2018). The selection indicator function $s_i$ is defined as:

$$s_i = \begin{cases} 1, & \text{if } \ell\left(f_\theta(\mathbf{x}_i), \tilde{y}_i\right) \le \tau, \\ 0, & \text{otherwise}, \end{cases} \tag{12}$$

where $\tau$ is a dynamically adjusted threshold. The training objective becomes:

$$\min_\theta \frac{1}{\sum_{i=1}^{n} s_i} \sum_{i=1}^{n} s_i \ell\left(f_\theta(\mathbf{x}_i), \tilde{y}_i\right). \tag{13}$$

A common approach is the small-loss trick, by focusing on low-loss samples, the model is less influenced by potentially mislabeled data. Methods like Co-teaching (Han et al., 2018), Co-teaching+ (Yu et al., 2019), JoCoR (Wei et al., 2020), and Co-learning (Tan et al., 2021) utilize two networks trained in parallel that teach each other using reliable samples. SELF (Nguyen et al., 2019) identifies clean samples by checking the consistency between network predictions and given labels, while

DivideMix (Li et al., 2020) employs a two-component mixture model to separate the training data into clean and noisy groups. Moreover, ELR (Liu et al., 2020) avoid overfitting to noisy labels by relying on early-learning.

**Learning dynamics reaearch for sample selection.** The intriguing generalization ability of modern DNNs has motivated extensive studies on their learning dynamics, which in turn has inspired a series of sample selection criteria using in Eq.(12) based on these dynamics. Studies have revealed that hard and mislabeled examples are typically learned during the later stages of training (Arpit et al., 2017; Song et al., 2019b, 2021; Maini et al., 2022; Li et al., 2023; Lin et al., 2024a; Hong et al., 2024). This empirical observation has led to the development of various training-time metrics to quantify the "hardness" of examples, such as learning speed (Maini et al., 2022; Jiang et al., 2021) and gradient variance. For instance, Agarwal et al. (2022) proposed *Variance of Gradients (VoG)* to estimate sample difficulty based on the temporal variability of gradient norms, while Novak et al. (2018) analyzed generalization through input-output Jacobian norms, connecting sensitivity in input space to learning robustness. These metrics have inspired LNL approaches that leverage learning dynamics to select clean samples. Methods like Self-Filtering (Wei et al., 2022), FSLT & SSFT (Maini et al., 2022), SELFIE (Song et al., 2019a), and RoCL (Zhou et al., 2021) adopt criteria to identify clean samples based on their learning dynamics. The success of learning dynamics-based sample selection criteria in identifying high-confidence clean samples has driven researchers to further refine these strategies. By identifying a larger subset of clean samples for model training, the generalization performance of the trained model can be improved. (Xia et al., 2021) discovered that using loss alone to select CHEs is suboptimal. RLM (Li et al., 2024) obtain robust loss estimation for noisy samples.

An advanced paradigm for sample selection involves a positive feedback loop: iteratively optimizing the classifier and updating the training set. Under this loop, the model's performance gradually improves, leading to better sample selection capabilities and, consequently, an enhanced ability to select clean hard examples. Me-Momentum (Bai and Liu, 2021) and Late Stopping (Yuan et al., 2023) employ similar positive feedback loops to iteratively update the model parameters and the training set, gradually improving the model's performance on noisy data.

**Hard label noise.** Various forms of *hard label noise* have been studied (Lin et al., 2023), including asymmetric noise (Scott et al., 2013), instance-dependent noise (Xia et al., 2020b), natural noise (Wei et al., 2021b), adversarially crafted labels (Zhang et al., 2024a), open-set noise (Wei et al., 2021a), and subclass-dominant noise (Bai et al., 2023). These noise types are designed from the perspective of the labels, aiming to simulate challenging real-world scenarios or malicious attacks. Recent work has also explored the impact of label noise in specific data distributions. For instance, H2E (Yi et al., 2022) and TABASCO (Lu et al., 2023) focus on the challenges posed by label noise in long-tailed distributions, where minority classes are more susceptible to mislabeling. NoiseCluster (Bai et al., 2023) introduces the concept of subclass-dominant label noise, where mislabeled examples dominate at least one subclass, leading to suboptimal classifier performance.

**Our contributions.** In contrast to prior studies that mainly focus on different types of label noise or sample selection based on learning dynamics, our work offers a fresh perspective by re-examining sample selection methods that rely on a model's early learning stages. We demonstrate that some samples hidden among those considered "confident" are, in fact, the most harmful when mislabeled. Specifically, we systematically investigate the detrimental impact of *Mislabeled Easy Examples (MEEs)*—mislabeled samples that are correctly predicted by the model early in the training process. This insight challenges the conventional assumptions of existing methods, which often prioritize samples learned early in training as being clean. Our findings highlight the need for a more cautious approach when selecting samples based on early learning confidence. By adopting a refined sample selection criterion that accounts for the potential harm of MEEs, we can seamlessly integrate this approach with existing sample selection method (Yuan et al., 2023) to further boost it performance. Furthermore, our proposed method conceptually linking to Sharpness-Aware Minimization (SAM) (Foret et al., 2021), but applied in the input space to identify stable yet mislearned examples.

# B  Quantitative Analysis of MEE Harm with Influence Functions

To provide a more direct and quantitative measure of the harm caused by Mislabeled Easy Examples (MEEs), we conducted an analysis using influence functions (Koh and Liang, 2017). This method allows us to calculate the impact score of different training samples on the model's performance (i.e., loss) on a clean, held-out validation set. A positive score indicates a harmful influence, while a negative score indicates a beneficial one. This analysis, performed using the `pytorch_influence_functions` library on CIFAR-10 with 40% instance-dependent noise, parallels the experiment in Figure 2. We compared three distinct sample categories: (1) MEEs, defined as the first 4,000 mislabeled samples learned during training; (2) Clean Easy Examples, the first 4,000 clean samples learned; and (3) Mislabeled Hard Examples, the last 4,000 mislabeled samples learned. The results provide direct, quantitative proof of our central claim. While all mislabeled examples exhibited a harmful positive influence, the average harm of MEEs was significantly greater than that of Mislabeled Hard Examples, with a mean influence score of 4.96 versus 3.01 (and a median of 4.48 vs. 3.26). This demonstrates at a microscopic level that MEEs are far more detrimental to the model. In contrast, the Clean Easy Examples showed a mean negative influence score of -2.22 (median -1.89), confirming their beneficial impact on generalization. This analysis further substantiates our claim in Section 2.2 regarding the unique and disproportionate harm caused by MEEs.

# C  Discussion on Broader Applicability

While the empirical validation in this paper is focused on visual classification, the core concepts of Mislabeled Easy Examples (MEEs) and Early Cutting are not inherently limited to this domain. We briefly discuss the potential translation of these ideas to other areas. In Natural Language Processing (NLP), an MEE could manifest as a text sample with strong but misleading keywords. For instance, a sarcastic positive review such as, "Wow, I can't believe how awful the service was," being mislabeled as "Positive". A model might quickly and confidently learn this incorrect association due to the presence of words like "Wow" and "can't believe", which often appear in positive contexts. The Early Cutting criteria could be conceptually adapted by, for example, measuring the gradient norm with respect to the input word embeddings to gauge stability. Similarly, in regression tasks, an MEE might be a data point where a specific feature has a strong, simple, and local linear relationship with the target value, but this relationship is globally spurious or an artifact of noise. The model would likely fit this deceptive pattern early in training. Our criteria could be conceptually translated by identifying samples that the later-stage model predicts with high confidence (e.g., in a value range far from the given label, thus incurring high loss) and stable input gradients. Though these examples are conceptual, they suggest that the underlying principle of identifying and removing confidently mislearned easy examples (MEEs) could be a generalizable strategy for enhancing model robustness across diverse machine learning domains.

# D  Detailed Settings

## D.1  Datasets

*CIFAR-10* and *CIFAR-100* (Krizhevsky et al., 2009) are standard image classification datasets consisting of $32 \times 32$ color images. Both datasets were divided into 50,000 training images and 10,000 test images. *CIFAR-N* (Wei et al., 2021b) is a version of CIFAR-10 and CIFAR-100 with real-world noisy labels collected from Amazon Mechanical Turk. These datasets simulate real-world scenarios where labels may be noisy due to human error. We used a consistent 90%-10% data splits for training and validation across runs in all competitors.

*WebVision* (Li et al., 2017) is a large-scale dataset containing over 2.4 million web images crawled from the internet. It covers the same 1,000 classes as the *ILSVRC12 ImageNet-1K* dataset (Deng et al., 2009) but includes noisy labels due to the automatic collection process. *ILSVRC12 ImageNet-1K* (Deng et al., 2009) is a large-scale dataset of natural images with 1,000 classes. We used it to assess the scalability of our method on real-world data with synthetic noise.

## D.2   Noise Settings

In preliminary presentation of our proposed method's effectiveness (Table 1), we tested four types of synthetic label noise. For *Symmetric Noise*, each label has a fixed probability $r$ of being uniformly flipped to any other class. *Asymmetric Noise* flips labels to similar but incorrect classes, mimicking mistakes that might occur in real-world classification tasks. *Pairflip Noise* involves flipping labels to a specific incorrect class in a pairwise manner. *Instance-Dependent Noise* (Xia et al., 2020b) is a more challenging setting where the probability of label corruption depends on the instance features. It reflects more realistic scenarios where difficult or ambiguous examples are more likely to be mislabeled.

Following prior practices (Bai and Liu, 2021; Yuan et al., 2023), we primarily focused on *Symmetric* and *Instance-Dependent* noise types in our baseline comparisons (Table 2 and 3), as they are the most common and challenging synthetic noise settings used to evaluate robustness methods. We experimented with noise rates of 20% and 40% to assess our method's performance under varying noise intensities. For the *CIFAR-N* task, we utilized the provided noisy labels.

## D.3   Model Architectures

We employed variants of the ResNet architecture (He et al., 2016) in all our experiments, training each model from scratch. Specifically, we used *ResNet-18* for *CIFAR-10*, *ResNet-34* for *CIFAR-100*, and *ResNet-50* for *WebVision* and *ImageNet-1K* datasets. This selection aligns with previous works and provides appropriate model capacity relative to each dataset.

## D.4   Training Procedures and Hyperparameters

Training was performed using *Stochastic Gradient Descent (SGD)* with a momentum of 0.9 and a weight decay of $5 \times 10^{-4}$. The initial learning rate was set to 0.1 and decayed using a cosine annealing schedule without restarts, decreasing to $1 \times 10^{-5}$ over the course of training. The number of training epochs was set to 300 for *CIFAR*, 200 for *WebVision*, and 150 for full *ImageNet-1K* experiments. Batch sizes were set to 32 for *CIFAR* datasets and *WebVision*, and 256 for *ImageNet-1K*.

To enhance the robustness of our sample selection model, we also incorporated certain strategies from prior works (Lin et al., 2024b; Li et al., 2024), training two networks and each network learn from the other's soft predictions and utilizing exponential moving averages to stabilize training. Weak data augmentation techniques were applied during training to improve generalization. These included random cropping with a padding of 4 pixels, random horizontal flipping, and normalization using the dataset-specific mean and standard deviation.

## D.5   Sample Selection Mechanism

Building upon the *Late Stopping* strategy (Yuan et al., 2023), we iteratively select a confident subset $\mathcal{D}^s$ of training samples, progressively reducing mislabeled data and enhancing the model's focus on clean samples. We identify early-learned samples based on their *learning times*. For each sample $(\mathbf{x}_i, \tilde{y}_i)$, we define its learning time $LT_i$ as the earliest epoch when the model's prediction stabilizes:

$$LT_i = \min \left\{ T_i \mid \hat{y}_i^{E_i-2} = \hat{y}_i^{E_i-1} = \hat{y}_i^{E_i} = \tilde{y}_i \right\}, \tag{14}$$

where $\hat{y}_i^t$ denotes the model's predicted label at epoch $e$.

To further address the issue of *Mislabeled Easy Examples* (MEEs), we introduce an *Early Cutting* step in the training loop. We first select candidates using an *Early Cutting Rate* $\gamma$ of 1.5, which corresponds to selecting the earliest $\approx \frac{2}{3}$ of samples learned. Within these candidates, we remove samples that meet all three of the following criteria (detailed in Section 3). First, we consider samples with high loss, specifically those within the top 10% of loss values $L_i$. Second, we look at samples with high prediction confidence, namely those within the top 20% of confidence scores $c_i$. Third, we identify samples with low gradient norms, that is, those within the bottom 20% of gradient norms $g_i$. By removing samples that satisfy all three conditions, we aim to eliminate MEEs that the model has confidently mislearned early on.

The refined subset $\mathcal{D}'^s$ is then used for subsequent training. We repeat the sample selection process for a total of $I_{\text{rate}}$ rounds (set to 3), progressively improving data quality and model performance. The

proportion of $\mathcal{D}^s$ retained in each round is calculated to achieve an overall retention rate equal to the complement of the noise rate after $I_{\text{rate}}$ rounds. For example, with a noise rate of 40% (aiming to retain 60% of the data), the retention rate per round is $(60\%)^{1/3} \approx 84\%$.

## D.6 Baselines and Competitors

We re-implemented these methods under the same experimental settings as our proposed method. When re-implementing CSGN (Lin et al., 2024b) using only supervised learning for Table 2, 3, 4, and 5. We used the AdamW (Loshchilov and Hutter, 2019) optimizer and a stepped decay learning rate schedule, as specified in the original code. Notably, CSGN (Lin et al., 2024b) cannot handle tasks with too many classes such as ImageNet-1k well.

## D.7 Training Time and Computational Complexity

**Computational Complexity.** The additional computational overhead introduced by the Early Cutting step itself, when applied to a subset of $n$ samples, involves three main operations:

1. Vectorized computation of the three metrics (loss $L_i$, confidence $c_i$, and gradient norm $g_i$): This is achieved in $O(n)$ time.

2. Identification of percentile thresholds: This requires sorting operations on the metrics, which takes $O(n \log n)$ time.

3. Filtering the samples based on these thresholds: This is an $O(n)$ operation.

Thus, the dominant term for the Early Cutting step is $O(n \log n)$ per application. Crucially, Early Cutting is executed only once per training round (e.g., after a certain number of epochs leading to a model state $f_{\theta^t}$), rather than per epoch. This significantly amortizes its cost over the entire training process. For instance, in our CIFAR-10 experiments using a ResNet-18 architecture, a single training epoch typically required approximately 42.7 seconds. The cumulative overhead for all Early Cutting operations throughout the entire training (e.g., 3 rounds) was approximately 70.3 seconds. This represents less than 1% of the total computational cost for a typical 200-epoch training process.

Taking a more demanding scenario like CIFAR-100 with ResNet-34 as example, where the full Early Cutting method takes approximately 15 hours, the contribution of the Early Cutting specific steps remains proportionally small. The majority of the time is consumed by the base iterative sample selection and model retraining process. Table 7 provides an illustrative breakdown.

Table 7: Illustrative runtime breakdown for "Early Cutting (Original)" on CIFAR-100/ResNet-34.

| Component | Illustrative Runtime Contribution |
|---|---|
| Base sample selection & retraining (iterative) | ~15.2 hours |
| *Early Cutting (Ours)* MEEs filter (cumulative) | ~0.1 hours |
| Total Training Time | ~15.3 hours |

**Overall Training Hours and Variants of Early Cutting:** While the Early Cutting step itself is efficient, the overall training duration also depends on the number of iterative refinement rounds and the base training time per round inherited from the underlying iterative sample selection framework (e.g., based on Late Stopping). We explored faster variant of our Early Cutting framework, *Early Cutting (Faster)*, which performed only one sample selection iteration (compared to three iterations in our original design), to balance performance with computational budget. Table 8 provides a comparative overview of total training hours and performance for various methods and our Early Cutting variants on CIFAR-100 with ResNet-34, tested on a single NVIDIA RTX 4090 GPU.

## D.8 Ablation Study on Each Component in Early Cutting Method

To validate the contribution of each distinct component and the design choices within our Early Cutting strategy, we conducted an ablation study. The Early Cutting method identifies Mislabeled Easy Examples (MEEs) based on their characteristics at a later training stage $t$: high loss $L_i$, high

Table 8: Comparison of total training hours and test accuracy (mean±std, CIFAR-100, ResNet-34, 40% Symmetric noise rate).

| Method | Runtime (Hours) | Test Accuracy (%) |
| --- | --- | --- |
| Me-Momentum (Bai and Liu, 2021) | ~15 | 63.99 ±0.56 |
| Late Stopping (Yuan et al., 2023) | ~17 | 61.71 ± 0.25 |
| RLM (Li et al., 2024) | ~4 | 67.31 ± 0.64 |
| CSGN (Lin et al., 2024b) | ~9 | 65.43 ± 0.52 |
| Early Cutting (Faster - Ours) | ~9 | 69.53 ± 0.10 |
| Early Cutting (Original - Ours) | ~15 | 69.94 ± 0.30 |

prediction confidence $c_i$, and low gradient norm $g_i$. Additionally, the 'early cutting rate' itself, which determines the initial pool of early-learned samples considered for MEE filtering, is a key aspect.

We performed the ablation study on a benchmark dataset (e.g., CIFAR-10/100 with a specific noise setting, for which the provided results are shown below, assumed to be for a representative scenario like CIFAR-10 with 40% instance noise or similar, leading to the baseline 'No Early Cutting' accuracy of 83.12%). The results, detailed in Table 9, quantify the impact of removing each component or not applying Early Cutting at all.

Table 9: Ablation study results demonstrating the contribution of each component in Early Cutting.

| Method / Variant | Test Accuracy |
| --- | --- |
| Full Early Cutting | 84.57% |
| - Early cutting rate | 84.12% |
| - Loss values criterion | 82.36% |
| - Confidence scores criterion | 84.10% |
| - Gradient norms criterion | 84.07% |
| No Early Cutting (Baseline using $\mathcal{D}^s$ only) | 83.12% |

The results demonstrate that:

- The full Early Cutting method (84.57%) significantly outperforms the baseline where no MEE filtering is applied to the initially selected confident subset (83.12%). This highlights the substantial benefit of the recalibration and MEE removal process.

- Removing the loss values criterion causes the most significant drop in performance (to 82.36%), underscoring its critical role in identifying samples where the model's later-stage understanding contradicts the noisy label.

- Omitting the early cutting rate consideration (which might imply either considering all of $\mathcal{D}^s$ for MEE filtering without pre-selection by earliness, or a suboptimal rate) leads to a noticeable decrease (84.12%), suggesting that focusing the MEE search on the very earliest learned examples is beneficial.

- Removing the confidence scores (84.10%) or gradient norms (84.07%) criteria also results in reduced accuracy, confirming their importance in refining the MEE identification by ensuring the model is certain and stable in its (incorrect) predictions for these MEEs.

These findings collectively validate that all components of the Early Cutting strategy synergistically contribute to its effectiveness in improving sample selection quality and model performance by precisely targeting and removing MEEs. The redundancy of clean, easy samples (learned early) also means that the accidental removal of a few such samples during this stringent filtering has a less detrimental impact compared to retaining harmful MEEs.

### D.9 Transferability of Default Parameters and Method Robustness

**Robustness to Threshold Variation.** Our method primarily relies on identifying proportions of samples based on their relative rankings for loss, confidence, and gradient norm. This percentile-based approach naturally adapts to different data distributions without requiring absolute threshold values.

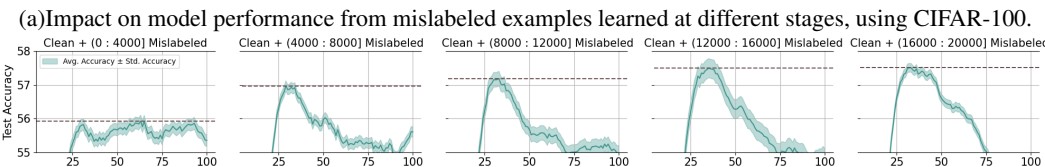

(a)Impact on model performance from mislabeled examples learned at different stages, using CIFAR-100.

Figure 6: Impact of mislabeled samples learned at different stages on model generalization performance.

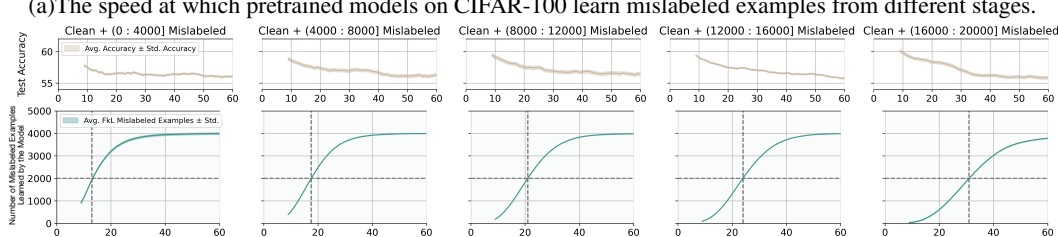

(a)The speed at which pretrained models on CIFAR-100 learn mislabeled examples from different stages.

Figure 7: Comparison of how pretrained models learn mislabeled examples from different learning stages.

As demonstrated in Figure 5 (in the main paper), varying these percentile thresholds by considerable factors (e.g., from 1/4 to 4 times the default proportions) results in less than a 1% change in test accuracy across various datasets. This low sensitivity is partly because our parameters are guided by the theoretical characteristics of MEEs: high loss (reflecting incorrect predictions by the more mature model), high confidence (model's certainty in these incorrect predictions), and low gradient norm (stability of these incorrect predictions). Furthermore, as clean samples learned early are often abundant and exhibit redundancy, the mistaken removal of a small fraction of these due to slight variations in thresholds does not significantly impair generalization.

**Consistent Performance with Default Hyperparameters.** As evidenced in Tables 2-5 (in the main paper), we applied consistent default hyperparameters for Early Cutting across a wide range of experimental setups. This includes different datasets (CIFAR-10, CIFAR-100, WebVision, full ImageNet-1k) and various noise conditions (symmetric, instance-dependent, real-world CIFAR-N). Despite this, Early Cutting consistently achieved state-of-the-art or highly competitive performance.

**Further Validation on New Datasets and Architectures.** To further substantiate the transferability and robustness of Early Cutting with its default settings, we conducted additional experiments on new datasets and with different model architectures, beyond those in the main paper. The results are presented in Table 10.

Table 10: Transferability of Early Cutting with default parameters to new datasets and architectures.

| Dataset | Model | Method | Test Accuracy |
|---|---|---|---|
| CIFAR-10 (Instance. 40%) | TinyViT | Early Cutting (Ours) | 75.42% |
| | | Late Stopping (Yuan et al., 2023) | 72.96% |
| | | CE (Cross-Entropy) | 69.31% |
| Fashion-MNIST (Instance. 40%) | ResNet-18 | Early Cutting (Ours) | 94.11% |
| | | Late Stopping (Yuan et al., 2023) | 92.81% |
| | | CE (Cross-Entropy) | 90.87% |

These results demonstrate that Early Cutting maintains its performance advantage even when applied to the transformer-based Tiny-ViT architecture on CIFAR-10, and on a different dataset like Fashion-MNIST with a ResNet-18 backbone, without any re-tuning of its core MEE identification parameters.

# E Evolution of Feature Space and Distance Ratios

To supplement the analysis in the main text (Figure 4(a)), which shows the feature space and distance ratios at an early stage (epoch 10), we provide additional visualizations in Figure 8 for later stages of training. These figures illustrate the evolution of the feature space and, crucially, how the model's representation of MEEs and other mislabeled samples changes over time.

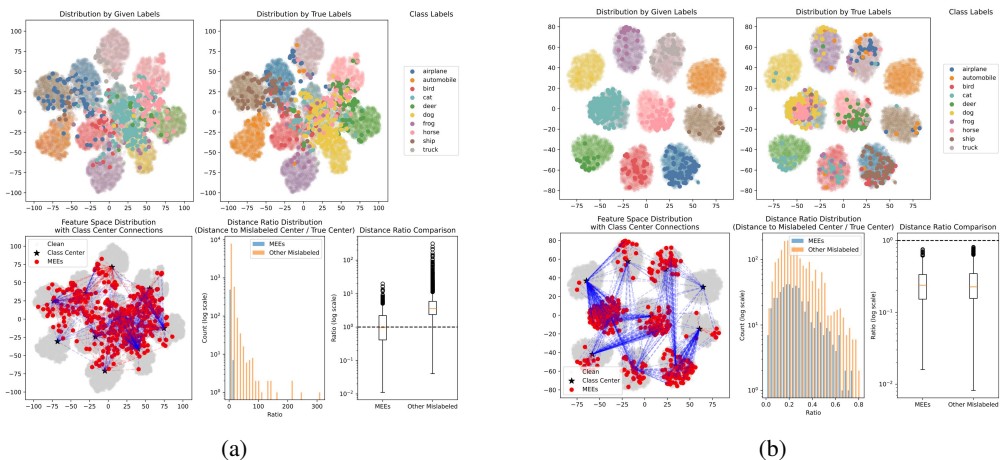

(a)                                                        (b)

Figure 8: Evolution of the t-SNE feature space visualization on CIFAR-10 (20% instance-dependent label noise) at (a) Epoch 40 and (b) Epoch 160. These figures complement Figure 4(a) (Epoch 10) from the main text.

