# OpenReview forum: "Enhancing Sample Selection Against Label Noise by Cutting Mislabeled Easy Examples"
_NeurIPS.cc/2025/Conference — NeurIPS 2025 poster_

### Official Review · Reviewer_Dz13 · 2025-06-19

**Clarity:** 3
**Significance:** 3
**Originality:** 3
**Rating:** 4
**Confidence:** 5

**Summary:**

This paper introduces a sample selection strategy called Early Cutting, aimed at improving robustness against label noise by identifying and eliminating a specific group of samples termed Mislabeled Easy Examples (MEEs)—i.e., mislabeled data points that are learned early and confidently by the model. The method leverages the model state at a later training epoch to re-evaluate these "early-learned confident samples", applying a joint criterion of high loss, high confidence, and low gradient norm to isolate and remove MEEs. Experimental results on CIFAR-10, CIFAR-100, WebVision, and full ImageNet-1K demonstrate promising performance improvements over state-of-the-art methods.

**Questions:**

1. Can the authors clearly distinguish how MEEs differ from early-learned noisy samples in Late Stopping (Yuan et al., ICCV 2023)? Are they essentially the same, and if not, what distinguishes them?

2. Why does the gradient norm reliably filter out MEEs rather than hard clean samples? Could the authors provide further evidence or ablations to justify this?

3. How sensitive is the method to the percentile thresholds (e.g., top 10% loss, bottom 20% gradient)? While a small ablation is given, it could be expanded.

4. Could the authors include comparisons against Me-Momentum and other dynamics-based selection methods targeting early-stage noise?

5. In the introduction, the authors state:
"Specifically, even with a low noise rate in the selected subset, the presence of certain mislabeled samples can still significantly impair the model’s generalization performance."
However, it is unclear what is meant by “low noise rate” in quantitative terms. Have the authors conducted any analysis to determine how low the noise rate needs to be for the impact on generalization to become negligible? In the experiments (e.g., Section 2.1), each subset contains 4,000 mislabeled samples among 34,000 total, which is still a relatively high proportion (~11.7%).
Would the proposed conclusions still hold in more extreme settings—for instance, if the number of mislabeled examples was fewer than 1,000 among 30,000+ clean samples (i.e., <3%)? A discussion or additional experiment under such conditions would help validate the generality of the claim.

6. When integrating the proposed Early Cutting method into the MixMatch semi-supervised learning framework (as described in Section 4.3), it is unclear whether the labeled and unlabeled subsets are fixed after the initial selection, or dynamically updated at each epoch during training.
Could the authors clarify whether the confident (labeled) and unconfident (unlabeled) sets are re-evaluated iteratively, and if so, how often the update occurs? This detail is crucial for understanding the training dynamics and the consistency regularization effect.

**Ethical Concerns:**

["NO or VERY MINOR ethics concerns only"]

**Final Justification:**

Most of my concerns have been addressed, and I am willing to raise my score to 4 (Borderline Accept). I sincerely thank the reviewer for the detailed response, as well as the other reviewers, AC, PC, and the organizing committee for their hard work.

**Limitations:**

The paper would benefit from a more thorough discussion of the limitations regarding the generality of its claims. In particular, the authors argue that even with a low noise rate, harmful mislabeled samples (MEEs) can significantly degrade generalization. However, no quantitative analysis is provided to determine how low the noise rate must be for the harmful effects to become negligible. For instance, if only 500 or 1000 mislabeled samples are mixed into 30,000+ clean examples (i.e., <3%), would the same degradation still occur?
In addition, the Early Cutting method may mistakenly eliminate a small number of clean samples—especially borderline or ambiguous cases—due to its reliance on high-loss, high-confidence, and low-gradient criteria. While the authors argue that such early-learned samples are redundant, this might not hold in low-data or long-tail scenarios where each clean instance carries significant value. A more balanced discussion of this trade-off would improve the paper's robustness.

**Paper Formatting Concerns:**

There are no major formatting issues in the paper.

**Quality:**

2

**Strengths And Weaknesses:**

Strengths: \
1.The paper highlights an overlooked yet important issue: not all noisy labels are equally harmful, and some “easy” mislabeled examples can be particularly misleading.\
2.Experimental results are extensive and evaluated across diverse datasets and noise settings.\
3.The method is simple yet principled, with clearly defined criteria (loss, confidence, gradient norm).\
4.The authors provide visual and empirical evidence to support the existence and impact of MEEs (e.g., Figure 1, Figure 4).\

Weaknesses: \
1.Overstatement of Novelty / Lack of Clear Contribution Boundaries\
The authors introduce the concept of Mislabeled Easy Examples (MEEs) and claim that these are particularly harmful mislabeled samples that are confidently learned by the model early in training. They state (e.g., Line 61):
“We discover that mislabeled samples correctly predicted by the model early in training disproportionately harm model’s performance…”
However, this phenomenon has been studied in prior works, including:
a. Me-Momentum (Bai & Liu, CVPR 2021), which proposes a momentum-based mechanism to identify memorized noisy samples early in training.\
b. Late Stopping (Yuan et al., ICCV 2023), which explicitly analyzes the impact of early-learned noisy examples and designs a strategy to mitigate their negative effect.\
Although these works are briefly acknowledged in Appendix A, they are not properly discussed or contrasted in the main text—particularly in Sections 2.2 and 3 where MEEs and the proposed Early Cutting strategy are introduced. The current manuscript presents the identification of MEEs as a novel discovery without clearly delineating the incremental contributions beyond these prior works.
The authors should explicitly cite and compare these related works in the main body and clearly articulate the difference between MEEs and previously defined early-learned noisy examples. If the contribution lies in improved detection or filtering of such examples (e.g., via gradient stability), that should be clearly emphasized as the novel component. Without this clarification, the claim of novelty appears overstated.\
2. Ambiguity in Method Motivation\
The use of the gradient norm to distinguish MEEs from hard samples (Equation 5) lacks solid theoretical justification. Why does low gradient norm necessarily imply "false patterns memorized"? The connection is heuristic and may not generalize.
See Lines 200–204: “MEEs tend to have low gradient norms because the model has confidently mislearned them…”
This needs more analysis or ablation to support.\
3. Lack of Baseline Comparisons\
While experiments include many methods, the paper omits direct comparisons with methods that also target early-learned noisy samples—e.g., “Late Stopping” or “Me-Momentum”，which are representative state-of-the-art methods targeting similar early-stopping-based noisy label issues. This omission makes it difficult to convincingly demonstrate that the proposed method outperforms existing approaches in the same problem setting.\
4. Limited Novelty Beyond Gradient-Based Criterion\
If we set aside the motivational premise—i.e., that early-learned mislabeled samples are especially harmful—since this has already been discussed in prior work (e.g., Me-Momentum, Late Stopping), then the core novelty of this paper is mainly the incorporation of the gradient norm as an additional filtering criterion. However, the use of combined loss and confidence thresholds for filtering mislabeled data is not new and has been explored in various existing methods. Therefore, the paper's overall innovation may appear limited unless the unique contribution of the gradient-based criterion is more clearly justified and empirically highlighted.

---

> ### Author Rebuttal · Authors · 2025-07-28
>
> Dear Reviewer Dz13 and Area Chair,
>
> We sincerely thank you for your valuable time in reviewing our manuscript. We have carefully studied all your questions and will address each point below.
>
> >**On the Paper's Core Contribution and Baseline Comparisons**
>
> **We would like to respectfully clarify a key point regarding the baseline comparisons.** The reviewer noted that our work was missing comparisons to Late Stopping [1] and Me-Momentum [2] (Weakness 3). We would like to respectfully point out that **both of these methods were included as core baselines in our main experimental results (Tables 2, 3)**. The results demonstrate that our proposed method significantly outperforms them across all tests.
>
> **This point is directly related to our core contribution, which we also wish to clarify.** The reviewer suggested that our motivation (that early-learned noisy samples are particularly harmful) has been discussed by Late Stopping [1] and Me-Momentum [2] (Weakness 1). We respectfully argue that **these paper are not discuss this motivation**.
>
> Specifically, methods like Late Stopping [1] and Me-Momentum [2] are built on a core assumption: that models learn simple, clean samples first, and therefore, examples learned early in training are trustworthy. Our paper's central contribution is to demonstrate, through empirical analysis, that this assumption is risky and can even be detrimental. Our novelty is three-fold:
>
> * 1.  We are the first to systematically identify and define the concept of *Mislabeled Easy Examples* (MEEs): We show that while these samples are "easily learned" early in training, they cause disproportionate harm to the model's generalization ability (as shown in Fig. 1 and Fig. 2).
> * 2.  We propose a counter-intuitive solution (Early Cutting): Based on our insights into MEEs, our novel Early Cutting method introduces a counter-intuitive calibration mechanism. It uses a later-stage, overfitted model to correct and purify the confident set selected during the early stage.
> * 3.  We provide a precise identification tool: To enable Early Cutting, we designed the combined criteria of high loss, high confidence, and low gradient norm. This is not a simple heuristic but a standard specifically engineered to pinpoint MEEs within a late-stage model.
>
> **Our work is not an incremental improvement on existing methods but rather a significant revision to a mainstream methodology in learning with noisy labels, for which we provide a solution.** We would like to revise the Introduction and Related Work sections to more clearly emphasize this critical distinction and our contribution.
>
> >**2. Justification for Using Gradient Norm to Identify MEEs (Response to Weaknesses 2, 4)**
>
> You raised an excellent question regarding why a low gradient norm can reliably help filter out MEEs.
>
> * Theoretical Motivation : When a model "memorizes" an incorrect pattern, it gives a wrong but high-confidence prediction for a sample (e.g., predicting an image of a plane labeled "ship" as "ship" with 99% probability). For this sample, the loss (relative to its incorrect label $y_i$) will be very high. But, for a small gradient norm indicates that the loss is insensitive to small perturbations in $\mathbf{x}_i$, suggesting a strong (but potentially incorrect) association between the input features and the predicted label. MEEs tend to have low gradient norms because the model has confidently mislearned them, making the loss stable even under input perturbations. (Section 3, line 202-206)
>
> * Empirical Validation : To validate that the gradient norm is a crucial component, we provided empirical evidence in our ablation study (Appendix B.9, line 623). We showed that removing the gradient norm criterion from our selection process leads to a significant drop in final model performance.
>
> >**3. Details on MixMatch**
>
> Thank you for pointing this out; this indeed requires clarification. In our current implementation, the confident set (labeled data) and un-confident set (unlabeled data) are determined once and remain fixed during the subsequent MixMatch training. We will add this explicit detail to Section 4.3 of the paper.
>
>
> >**4. Performance at Low Noise Rates**
>
>
> We thank you for this interesting question about whether our experimental findings generalize to scenes with extremely low noise rates. Following your constructive suggestion, we have conducted additional experiments to test our hypothesis in very low noise rate.
>
> To do this, we constructed two separate low-noise training sets for both CIFAR-10 and CIFAR-100. Each set contains the full clean dataset plus 2,000 mislabeled samples (~6% noise rate, which is much lower than the common low noise setting, 20% noise rate, in learning with noise labels). The difference lies in *which* mislabeled samples were chosen, based on their learning-time ranking from our original, higher-noise experiments (as detailed in Section 2).
> All other experimental settings were kept identical to those used for Figure 2. The results, shown in the table below, confirm that even at this very low noise rate, the mislabeled samples learned earliest by the model are significantly more harmful to final generalization performance:
>
> | Test Accuracy | Clean+(0:2000] Mislabled (easy mislabeled) | Clean+(18000:20000] Mislabled (hard mislabeled)
> | :--- | :---: | :---: |
> | **CIFAR-10** | 85.99% | 88.09% |
> | **CIFAR-100** | 57.74% | 60.10% | *
>
>
> >**5. Discussion of Trade-offs and Limitations (Response to Limitations)**
>
> You've raised an important point about the risk of our method removing a small number of clean samples (i.e., false positives). We agree this trade-off exists in any sample selection method. The critical question is whether the benefit of removing harmful mislabeled examples outweighs the cost of losing these false positives.
>
> Our Early Cutting method is designed to mitigate this risk through its strict, multi-part criterion. By engineering the method to require that a sample simultaneously meets three conditions (high loss, high confidence, and low gradient norm), we reduce the likelihood of removing hard clean samples. For instance, a challenging clean sample, such as from a long-tail class, might exhibit high loss. However, the model's confidence in its prediction would be low, and its gradient norm would be high, because a merely difficult sample does not confuse the model in the same way. Therefore, difficult long-tail samples typically do not fit our criteria for removal. Our experimental results confirm that this multi-part constraint significantly improves selection precision, ensuring the net effect is a substantial gain in performance.
>
> Despite these design choices, we agree with your sentiment that this trade-off becomes more critical in scenarios with extremely sparse data. We will add a *Section 6: Limitations* to the paper to discuss this trade-off, acknowledging that the method may require adjustments or further research for such specific use cases.
>
> ---
>
> Thank you once again for your thoughtful feedback.
>
> Best regards,
>
> The Authors
>
> ---
>
> Reference:
>
> [1] Late Stopping: Avoiding Confidently Learning from Mislabeled Examples, ICCV 2023.
>
> [2] Me-Momentum: Extracting Hard Confident Examples from Noisily Labeled Data, ICCV 2021.

---

> > ### Comment · Reviewer_Dz13 · 2025-08-05
> >
> > Thank you for your detailed response. Most of my concerns have been addressed, and I am willing to raise my score to **4** (Borderline accept).

---

> > > ### Author Response · Authors · 2025-08-06
> > >
> > > Dear Reviewer Dz13,
> > >
> > > Thank you for your detailed review and for your thoughtful engagement during the discussion period.
> > >
> > > We were very pleased to see that our detailed response addressed your concerns, and we sincerely appreciate you raising your score to positive in light of our discussion and revisions.
> > >
> > > Best regards,
> > >
> > > Authors

---

### Official Review · Reviewer_bRY1 · 2025-06-30

**Clarity:** 4
**Significance:** 4
**Originality:** 4
**Rating:** 5
**Confidence:** 4

**Summary:**

This paper shows that not all incorrect labels are equally bad for training models. Mislabeled data learned early, which the authors call Mislabeled Easy Examples (MEEs), are especially harmful. The paper introduces a method named Early Cutting to fix this issue. This technique uses a more mature model from later in the training process to find and remove these MEEs from the initially selected data. Experiments on datasets like CIFAR, WebVision, and ImageNet confirm that this strategy successfully improves model performance by filtering out these specific harmful examples.

**Questions:**

1. Please use different notations for time ($T_i$), learning time ($LT_{i}$), and the noise matrix ($T$).

2. Consider adding citations for fundamental methods like the SGD and AdamW optimizers you used.

3. Please clarify the “Early Cutting Rate” hyperparameter and its selection in the main paper.

**Ethical Concerns:**

["NO or VERY MINOR ethics concerns only"]

**Final Justification:**

Thank you for your response. It has addressed most of my concerns, and I will maintain my current rating.

**Limitations:**

yes

**Quality:**

4

**Strengths And Weaknesses:**

Strengths:
1. This paper identifies a new problem: mislabeled examples learned early during training cause disproportionate harm to the model.

2. This work proposes a counter-intuitive but effective method that uses a later model to correct the initial confident dataset.

3. The method is thoroughly validated with strong results across various datasets, noise types, and strong competitor methods.

Weaknesses:
1. The use of similar notations like “$T_i$” on Line 98, “$LT_{i}$” in Eq. (1), and the noise transition matrix “$T$” on Line 444 can easily confuse readers.

2. Citations for some important literature are missing, for example, the SGD and AdamW optimizers.

---

> ### Author Rebuttal · Authors · 2025-07-28
>
> Dear Reviewer bRY1 and Area Chair,
>
> We sincerely appreciate the valuable suggestions, which have been instrumental in improving the quality of our manuscript.
> We agree with all the points you raised and have revised the paper accordingly. Below, we detail the specific changes made:
>
> ---
>
> >**On the Confusion of Notation:**
>
> We thank you for pointing out the confusion with the overloading of the symbols. We have carefully revised the notation throughout the manuscript to ensure clarity. We now use $\mathbf{E}$ to denote the number of training epochs.
>
> >**On Missing Citations:**
>
> We appreciate your reminder to include citations for foundational methods. We have now added the appropriate references for SGD [1], AdamW [2], Weight Decay [3], and SGD with momentum [4]. These citations can be found in Section 4.2 (Datasets and implementation) and in Appendix B (Detailed Settings).
>
> >**On Clarification of a Hyperparameter:**
>
> Thank you for pointing out the omission of the specific value for the "Early Cutting Rate" hyperparameter in the main paper. We have now specified in Section 3 (line 209) that the Early Cutting Rate was set to 1.5. In line with our treatment of other hyperparameters, we have also included a sensitivity analysis for the Early Cutting Rate in Section 4.3 and a corresponding ablation study in Appendix B.9.
>
> ---
>
> Thank you once again for your consideration of our work and for your constructive guidance. We are confident that these revisions have enhanced the readability of our paper.
>
> Best regards,
>
> Authors
>
> ---
>
> References:
>
> [1] A stochastic approximation method, The annals of mathematical statistics, 1951.
>
> [2] Decoupled Weight Decay Regularization, ICLR 2019.
>
> [3] A Simple Weight Decay Can Improve Generalization, NeurIPS 1991.
>
> [4] Learning representations by back-propagating errors, Nature, 1986.

---

> > ### Comment · Reviewer_bRY1 · 2025-08-05
> >
> > Thanks for the author's reply! It has cleared up most of my worries. I will keep my rating the same.

---

> > > ### Author Response · Authors · 2025-08-05
> > >
> > > Dear Reviewer bRY1,
> > >
> > > Thank you for your support of our work and for maintaining your positive "Accept" rating after reviewing our response. We are glad that our rebuttal successfully addressed most of your questions.
> > >
> > > Your feedback on issues such as the inconsistent notation and missing citations was very helpful in improving the manuscript's formal presentation and readability. We have carefully revised a new version according to your suggestions.
> > >
> > > Thank you again for your valuable time and constructive feedback.
> > >
> > > Best regards,
> > >
> > > Authors

---

### Official Review · Reviewer_5oqq · 2025-07-01

**Clarity:** 1
**Significance:** 2
**Originality:** 2
**Rating:** 2
**Confidence:** 4

**Summary:**

Sample selection in learning with noisy labels typically identifies a confident subset of training
examples by reducing the noise rate within the selected data. However, such a strategy often overlooks
the fact that not all mislabeled examples are equally harmful to model performance. In contrast,
through numerical experiments, this manuscript aims to demonstrate that predicted mislabeled
examples in the early stage of training, referred to as Mislabeled Easy Examples (MEEs), can be
particularly detrimental to performance. Building on this observation, the authors propose a method
called Early Cutting to filter out MEEs. The method is evaluated on several benchmark datasets,
including CIFAR, WebVision, and the full ImageNet-1k, to demonstrate its effectiveness.

**Questions:**

Below, I list some of my questions in the order of their appearance in the manuscript, rather than by significance. Please note that the list is not exhaustive, but is intended to illustrate key concerns.

\begin{itemize}
    \item  [1.]  Lines 39–43:

The statement ``As shown in Figure 1(a), we demonstrate that mislabeled samples which are correctly predicted by the model early in the training process disproportionately degrade performance" lacks clarity. Specifically:

A. There is no description of how Figure 1 was generated. As such, it is  unconvincing to support the authors’ claim.

B. Clarify the meaning of ``mislabeled samples which are correctly predicted by the model...".


C. The phrase "early in the training process" needs clarification. How early is considered ``early"? Does this vary across training algorithms or settings?

D. The phrase ``disproportionately degrade performance" is vague. Disproportionate to what? And whose performance is being degraded?


\item [2.] Lines 40-43:


In the statement ``In our analysis (see Section 2.2), we find that MEEs are often closer to the centers of their mislabeled classes in the feature space of classifiers trained in the early stages", how are the ``centers” defined? In what sense are MEEs considered closer to these centers, e.g., in terms of Euclidean distance, cosine similarity, or other metric?



\item [3.] Lines 190-213:


   The key idea is to introduce a set of suspicious samples ${\cal S}$
    in (3) and its refined version ${\cal S}^\prime$ in (5). While the rationale of using the scale of the cross-entropy loss, the prediction confidence, and  the gradient stability  seems reasonable, some fundamental issues are not addressed or fully discussed:

   A. the predicted label $\hat y_i$  is determined by the conditional probability $p_i^{(k)}$, which essentially requires the information on the clean label $y_i$ (BTW, there seems a typo in the definition of $p_i^{(k)}$: should $y$  be $y_i$?).
   In the presence of label noise, the true label $y_i$ is typically unknown.
     Then how do we determine $L_i$ and $c_i$
   in order to construct ${\cal S}$ and ${\cal S}^\prime$?




   B. Assuming the issues in A is resolved, how do we  ensure that ${\cal S}$ in (3) is not an empty   set? The constraints $L_i > \delta$ and $c_i> \tau$ can be equivalent written as
   $$
   p_i^{(y_i)} < \exp(-\delta) \ \text{and} \ p_i^{(\hat y_i)}> \tau.
   $$
   To make ${\cal S}$ mathematically meaningful and practically useful, appropriate constrains should be imposed on the thresholds $\delta$ and $\tau$. This important aspect seems to be neglected.

    Although the author(s) state  "Specifically, for all settings, we target the top 10\% for loss, top 20\% for confidence, and bottom 20\% for gradient norm. These percentile-based selections are intentionally kept fixed across all experimental settings
   to underscore the method’s general applicability and robustness, thereby obviating the need for dateset-specific hyperparameter tuning", this treatment of the threshold values $\delta$ and
   $\tau$ appears overly simplistic and does not adequately account for the diversity and  complexity of real-world data.



\item [4.] Lines 445 (in Appendix A):

      A setting of instance-independent label noise is presented in (7). However, this key assumption does not appear to be clearly indicated in the main text. How
        does this type of label noise relate to the construction of
       ${\cal S}$ and ${\cal S}^\prime$ in Section 3? Shouldn't these sets be constructed
        based on the noisy labels and the cross-entropy loss defined in equation (10), rather than using $L_i$ as defined in Line 189?

\end{itemize}

**Ethical Concerns:**

["NO or VERY MINOR ethics concerns only"]

**Final Justification:**

Dear AC:

All the rebuttals have been carefully reviewed. While I appreciate the authors' effort in trying to improve the work and some rebuttals help clarify certain comments,  critical concerns do not seem to be fully addressed. While the work may be of its own value, I do not feel it's up to the level of NeurIPS. As a result, I do not think I'm ready to raise the initial score.

Thank you

**Limitations:**

They do not seem to make limitations clear. Addressing the questions or concerns above could help improve the clarity of the work.

**Quality:**

1

**Strengths And Weaknesses:**

Strengths:


    1. This manuscript explores an important and often overlooked issue in sample selection strategies for learning with label noise.

     2.  Specifically, it aims to distinguish the varying negative impacts of mislabeled examples on model performance, which is valuable.

    3. The experimental results support the intended merit of the proposed method.


Weaknesses:

    1. The proposed methods is merely evaluated by numerical experiments, and there is no theoretical guarantee.

    2. Some concepts or terms are introduced without sufficient motivation or justification regarding their rationale or effectiveness.

    3.  Some notations are inconsistently defined or used. For example, the number of classes is denoted by $K$ in the main text (Lines 184–185), but by $C$ in
Appendix A on Related Work.

    4. Certain key statements lack clarity.

    5. Repetitive descriptions seem to appear a bit too frequently throughout the manuscript. While repetition can help emphasize important points, its overuse may affect the overall conciseness.

---

> ### Author Rebuttal · Authors · 2025-07-28
>
> Dear Reviewer 5oqq and Area Chair,
>
> Thank you for your detailed and constructive feedback. We appreciate your recognition of our work's motivation, that it is "valuable to distinguish the varying negative impacts of mislabeled examples on model performance", and your assessment that this is an "important and often overlooked issue in sample selection strategies for learning with label noise".
>
> You have raised specific, valuable points regarding the paper's clarity and methodological details. In the following, we address your concerns and detail the significant revisions we have undertaken based on your suggestions.
>
> >**1. On Improving Clarity (in response to Weaknesses 2, 4 & Questions 1, 2)**
>
> **1.1. Clarification of Figure 1 (in response to Q1)**
>
> We thank you for your questions regarding the clarity of Figure 1 and offer the following clarifications:
>
> * **Q1.A (Experimental Setup):** The experimental setup for Figure 1(a) is detailed in Section 2.1 (in the context of Figure 2). This experiment demonstrates the unique harm of MEEs by comparing the performance degradation when adding an equal number of MEEs versus Mislabeled Hard Examples to a clean dataset. In the revised manuscript, we have added an explicit reference to Section 2.1 when Figure 1 is first mentioned and have enhanced the figure caption with a concise summary of the setup.
> * **Q1.B (Definition):** The phrase "mislabeled samples which are correctly predicted by the model" refers to instances where a sample is inherently mislabeled, yet the model has learned to classify it according to its given (incorrect) noisy label.
> * **Q1.C (Early in Training):** The sentence immediately following the one you quoted explains this. We define the process of the model first learning these initial four thousand mislabeled samples as occurring "early in the training process".
> * **Q1.D (Disproportionate Impact):** The "disproportionate" impact is precisely what is illustrated in Figure 1(a). A training set contaminated with Mislabeled Easy Examples yields a model with significantly worse performance than one contaminated with the same number of Mislabeled Hard Examples, demonstrating that the negative impact of noisy labels is not uniform.
>
> **1.2. Definition of Key Concepts (in response to Q2)**
>
> Regarding the definition of "class centers" (Q2), we specified in both our analysis in **Section 2.2** and the caption of **Figure 4** that a "center" is the **mean embedding vector** of all samples belonging to that class, and distance is measured using the **Euclidean distance**. We also note that when this concept was first introduced in the introduction, we had already included a reference pointing to the detailed description via "(see Section 2.2)".
>
> **1.3. Notation and Prose (in response to W3, W5)**
>
> We have conducted a thorough pass of the manuscript to unify all notation (e.g., consistently using $K$ for the number of classes), streamline the prose, and remove redundant descriptions.
>
> We respectfully point out that while we strive for precision, our goal in the **Section 1 Introduction** is to provide a clear and concise overview of the phenomena we observe. For this reason, certain granular details are necessarily elaborated upon in their respective, dedicated sections.
>
> >**2. On Clarifying Methodological Details (in response to Weaknesses 1, 3, 5 & Questions 3, 4)**
>
> **2.1. On the Methodology (in response to Q3.A)**
>
> You suggested that the calculation of the conditional probability $p_i^{(k)}$ requires the true label $y$. There may have been a misunderstanding here, which we are happy to clarify:
> 1.  The term $p_i^{(k)}$ is a direct output of the model's forward pass and **requires no label information**. In the expression $Pr(y=k|x_i;\theta^t)$, the symbol **$y$** represents a **random variable** for the class label, and **$y=k$** denotes the event that this random variable takes the value of class $k$.
> 2.  The calculation of the loss $L_i$, the predicted label $\hat{y}_i$, and the confidence $c_i$ do not depend on the unknown true label. They use either the **given (noisy) label $y_i$** (during backpropagation) or the model's own probabilistic output $p_i$.
>
> While we believe our notation is standard, we have revised this section (lines 184-190) in the manuscript to use more explicit notation (e.g., using **$Y$** to denote the random variable) to prevent any possible ambiguity.
>
> **2.2. On the Instance-Independent Noise Assumption (in response to Q4)**
>
> We wish to clarify that our method **does not depend on the instance-independent label noise assumption**. The discussion of this assumption in Appendix A (Equation 7) is part of a standard protocol for describing how synthetic noisy datasets are generated; **it describes the data generation process**, not a prerequisite for our methodology. In Section 3, our method operates directly on the observed dataset $\mathcal{D}^s$ with its given (noisy) labels $y_i$, irrespective of how the label noise was generated. Crucially, as shown in **Tables 2-5**, we have already validated our method's performance under various noise types, including synthetic symmetric, synthetic instance-dependent, and real-world label noise.
>
> **2.3. On Ensuring the Set $\mathcal{S}$ is Non-Empty (in response to Q3.B)**
>
> We agree with your suggestion that the definition should ensure the set $\mathcal{S}$ is non-empty. We have incorporated a formal discussion of the constraints on the thresholds into the manuscript and updated Algorithm 1 accordingly. We would also like to humbly point out that in practice, our use of percentile-based thresholds effectively prevents the sets $\mathcal{S}$ or $\mathcal{S}'$ from being empty.
>
> **2.4. On Threshold Design (in response to Q3.B)**
>
> You noted that using fixed percentile thresholds might be "overly simple". Our rationale for this design choice was to prioritize generality, robustness, and ease of use, thereby avoiding complex hyperparameters that require fine-tuning for each dataset. Our sensitivity analysis (Figure 5) demonstrates that the method's performance is highly stable across a wide range of threshold values, which empirically validates the robustness of this design.
>
> **2.5. On the Empirical Nature of the Work (in response to W1)**
>
> We acknowledge that our work is primarily an empirically-driven study. We believe its main contributions are nonetheless significant: (1) identifying and defining a novel and impactful problem (the disproportionate harm of MEEs); (2) proposing a novel, effective, and practical solution (Early Cutting); and (3) validating this solution with extensive experiments on large-scale benchmarks, including full ImageNet-1k. We are confident that such empirical contributions hold significant value, particularly in an application-driven field like machine learning.
>
> ---
>
> We hope that these clarifications and our accompanying revisions will address your concerns and significantly improve your opinion of our paper. Thank you once again for your tremendous effort and valuable feedback.
>
> Sincerely,
>
> The Authors

---

> ### Comment · Reviewer_5oqq · 2025-08-03
>
> The reviewer thanks the authors for their efforts in addressing the comments and criticisms on the initial submission. While the rebuttal clarifies several points,  key concerns remain, as outlined below.\\
> %-------------------------------------------------------------------
> \textbf{Regarding ``1.1. Clarification of Figure 1 (in response to Q1)"}\\
> While the authors responded with: ‘Q1.C (Early in Training): The sentence immediately following the one you quoted explains this. We define the process of the model first learning these initial four thousand mislabeled samples as occurring "early in the training process",’ this response does not fully address the original question: ‘C. The phrase "early in the training process" needs clarification. How early is considered "early"? Does this vary across training algorithms or settings?'\\
> %-------------------------------------------------------------------
> \textbf{Regarding ``1.2. Definition of Key Concepts (in response to Q2)"}\\
> The authors clarified that, in their analyses, a 'center' refers to the mean embedding vector of all samples belonging to a class, and that distance is measured using the Euclidean metric, with a reference provided. Accompanying questions arise: why was Euclidean distance chosen over other metrics to reflect the closeness of MEEs to these centers? Would using a different distance metric alter the conclusions? Are there any experiments available to demonstrate the sensitivity (or robustness) of the results to this choice?\\
> %-------------------------------------------------------------------
> \textbf{Regarding ``2.1. On the Methodology (in response to Q3.A)"}\\
> Thank you for clarifying that $y_i$ represents 'the given (noisy) label'. However, this explanation appears to conflict with the notation used in Appendix A, where the 'observed noisy labels' are denoted by $\tilde{y}_i$ and the true label is denoted by $y_i$ (or $y$), which also aligns with standard practice in the label noise literature. Please clarify this inconsistency.\\
> In addition, the descriptions in Appendix A do not align well with Lines 183–189 in the main text, due to inconsistent notation and the imposition of equation (7), which implicitly assumes instance-independent label noise.\\
> %-------------------------------------------------------------------
> \textbf{Regarding ``2.3. On Ensuring the Set ${\cal S}$
> is Non-Empty (in response to Q3.B)"}\\
> The reviewer thanks the authors for their response:
> 'We agree with your suggestion that the definition should ensure the set is non-empty. We have incorporated a formal discussion of the constraints on the thresholds into the manuscript and updated Algorithm 1 accordingly. We would also like to humbly point out that, in practice, our use of percentile-based thresholds effectively prevents the sets from being empty.'\\
> However, as a revised version of the manuscript does not appear to be available, and the authors have not provided details on how the critical comments under '[3.] Lines 190–213' in the initial report have been addressed, the reviewer currently has no basis for assessing whether the revisions are acceptable.\\
> %-------------------------------------------------------------------
> \textbf{Regarding ``2.4. On Threshold Design (in response to Q3.B)"}\\
> The authors' response on this point does not  fully address the initial concern that ‘this treatment of the threshold values $\delta$ and $\tau$ appears overly simplistic and does not adequately account for the diversity and complexity of real-world data.’ While the authors mention that their design choice uses fixed rates, this choice does not seem to be directly linked to the threshold values $\delta$ and $\tau$ in the imposed constraints.\\
> Furthermore, although you state that ‘Our sensitivity analysis (Figure 5) demonstrates that the method's performance is highly stable across a wide range of threshold values, which empirically validates the robustness of this design,’ this claim is not entirely convincing. A typical concern is that numerical studies alone cannot capture the full range of data characteristics encountered in real-world applications. Even if Figure 5 illustrates the intended message, it does not necessarily justify the generality of the proposed approach.
> \\
> %-------------------------------------------------------------------
> \textbf{Regarding ``2.5. On the Empirical Nature of the Work (in response to W1)"}\\
> The reviewer appreciates the authors’ acknowledgement that their work is primarily an empirically driven study. It would be informative to include a theoretical guarantee to enhance the rigor and generality of the proposed method. What are the main challenges preventing the establishment of such a guarantee?

---

> > ### Author Response · Authors · 2025-08-05
> > **Response to Reviewer 5oqq (Round 2, part1)**
> >
> > Dear Reviewer 5oqq and Area Chair,
> >
> > Thank you for your continued engagement with our manuscript and for providing additional detailed feedback. We appreciate the opportunity to further clarify our work.
> >
> > ---
> >
> > >**Regarding 1.1: Clarification of "Early in Training" (in response to Q1.C)**
> >
> > We appreciate you pushing for a more precise definition. To clarify, the term "early in training" has both an operational and a conceptual meaning in our work.
> >
> > Operational Definition (for the experiment): In the specific context of the experiment in Figure 1(a), "early" is operationally defined by the learning order. The "initial four thousand mislabeled samples" are the first 4,000 mislabeled examples to be learned by the model according to the Learning Time ($LT_i$) metric defined in Eq. (1). This is a data-driven way to identify the earliest learned cohort.
> >
> > Conceptual Definition: More broadly, "early in the training process" refers to the initial phase where a deep neural network learns simple, dominant patterns from the data, prior to the onset of significant memorization of noisy or complex atypical examples. This term is based on [1], and is widely used in research of learning with noisy labels. The precise number of epochs this phase lasts does indeed vary with architecture, dataset, and optimizer settings. This is precisely why dynamics-based methods like ours rely on observing the empirical learning order of individual samples rather than using a fixed epoch number.
> >
> > >**Regarding 1.2: Definition of Key Concepts (in response to Q2)**
> >
> > You asked why Euclidean distance was chosen for our analysis and whether other metrics would alter the conclusions.
> >
> > Our choice was guided by the principle of methodological consistency. Our qualitative analysis in Figure 4 relies on t-SNE to visualize the high-dimensional feature space. The standard t-SNE algorithm [2] operates by preserving local neighborhood structures, which are fundamentally defined based on Euclidean distances. To ensure our quantitative analysis is directly comparable to our visualization, we consistently used Euclidean distance as well. This approach ensures that what we visualize qualitatively is precisely what we are measuring quantitatively.
> >
> > While other metrics like Cosine Distance could be employed, we believe that for the primary goals of this work, to first identify an important, overlooked problem and demonstrate its existence, using a standard and well-motivated metric is sufficient. We agree that a deeper exploration into the effects of different metrics is an interesting direction for future work.
> >
> > >**Regarding 2.1: On the Methodology and Notation (in response to Q3.A)**
> >
> > Following your suggestion, to resolve notational inconsistency, we have performed a revision to unify the notation throughout the entire manuscript, adopting the convention as you suggested.
> >
> > Here are the specific changes we have made:
> >
> > 1.  Consistent Notation for Labels:
> >     We now consistently use $\tilde{y}_i$ to denote the observed noisy label for sample $i$ from the dataset.
> >     The symbol $y_i$ is now reserved only for the true, unobserved clean label.
> >
> > 2.  Revision of Section 3 (Methodology):
> >     The definition of the dataset (previously Line 183) is now written as: \mathcal{D}^s = \{(\mathbf{x}_{i}, \tilde{y}_{i}) \}_{i=1}^N (This formula can not render correctly in the OpenReview's markdown env.).
> >     The definition of the cross-entropy loss (previously Line 189), which is computed using the given noisy label, is now written as: $\ell_i = -\log p_i^{(\tilde{y}_i)}$.
> >     All related descriptions in Section 3 have been updated to reflect this consistent notation.
> >
> > 3.  Equation (7) provides the formal definition for the classic noise model. It is included in our Appendix A Related Work as part of a standard background review of the LNL field, intended only to provide context for the reader.

---

> > ### Author Response · Authors · 2025-08-05
> > **Response to Reviewer 5oqq (Round 2, part2)**
> >
> > >**Regarding 2.3: On Ensuring the Set ${\cal S}$ is Non-Empty (in response to Q3.B)**
> >
> > To clarify the revision we have implemented, we have added a sentence in Section 3 (line 194): Note that $\cal S$ may be empty if $\delta$ and $\tau$ are incompatible (e.g., $\delta > -\log(1-\tau)$). In practice, we use percentile thresholds, so each criterion selects a fixed proportion of samples; empirically the intersection is non-empty in all our experiments, and if $|\cal S|=0$ we simply skip removal for that round.
> >
> >
> > >**Regarding 2.4: On Threshold Design (in response to Q3.B)**
> >
> > We appreciate your perspective. Our primary goal was to propose a simple and practical solution to the newly identified problem of MEEs. The use of fixed percentile thresholds was a deliberate design choice to maximize this practicality, avoiding the need for complex, dataset-specific hyperparameter tuning.
> >
> > While we agree that numerical studies alone cannot capture the full range of all possible data characteristics, our claim of general applicability is supported by a methodology that aligns with established evaluation standards in the LNL community. The fact that this simple, fixed approach achieves strong results across a wide range of diverse and standard benchmarks (CIFAR, WebVision, full ImageNet) and noise types (symmetric, instance-dependent, real-world) demonstrates that it is already a highly effective and reliable solution. We believe this practical effectiveness, born from a simple design, is a core strength of our work that addresses the problem we set out to solve.
> >
> > >**Regarding 2.5: On the Empirical Nature of the Work (in response to W1)**
> >
> > We appreciate the acknowledgement and agree that a theoretical guarantee would be a valuable addition. The primary challenges, as you may know, are significant:
> > The training dynamics of deep networks involve highly non-convex optimization, which is notoriously difficult to analyze theoretically.
> > The very existence of MEEs is an empirical, data-dependent phenomenon, making it difficult to formulate the general assumptions required for a formal proof.
> >
> > Given these, we trust our work provides a complete contribution by first empirically identifying and characterizing an important, overlooked problem (MEEs), and then proposing a practical and demonstrably effective algorithmic solution (Early Cutting). This research paradigm, establishing a phenomenon and providing a practical solution, is consistent with many impactful papers in our field and, we humbly feel, constitutes a significant contribution in its own right.
> >
> > ---
> >
> > We appreciate this opportunity to refine the manuscript. Following your feedback, we have try our best to clarify your main concerns, especially the critical points that our method requires clean labels or assumes instance-independent noise. With these foundational issues now resolved, and with the additional details provided above, we sincerely hope this provides a solid basis for a positive re-evaluation of our work.
> >
> > Best regards,
> >
> > Authors
> >
> > ---
> >
> > Reference:
> >
> > [1] Arpit, Devansh, et al. "A closer look at memorization in deep networks." International conference on machine learning. PMLR, 2017.
> >
> > [2] Maaten, Laurens van der, and Geoffrey Hinton. "Visualizing data using t-SNE." Journal of machine learning research 9.Nov (2008): 2579-2605.

---

> > > ### Comment · Reviewer_5oqq · 2025-08-08
> > >
> > > This is to acknowledge that the second-round rebuttal has been carefully read and assessed. I thank the authors for their continued efforts to address the concerns raised in this submission.

---

> ### Author Response · Authors · 2025-08-08
>
> Dear Reviewer 5oqq,
>
> Thank you for your time and for acknowledging our response.
>
> We are glad we have clarify your initial key concerns. In particular, we believe the foundational misunderstandings regarding our method's core mechanics (e.g., that our method does not require clean labels, nor does it rely on the instance-independent noise assumption) have been fully resolved. Following your suggestions in the second round of discussion, we also addressed more detailed points, such as the specific strategy to ensure the set $\mathcal{S}$ is non-empty and the rationale for our choice of distance metric.
>
> We understand that your subsequent questions, such as the discussion on theoretical guarantees, are constructive suggestions aimed at further improving the work. We believe that with the above-mentioned fundamental issues clarified, the initial concerns that led to the very low scores for this work in Quality (1), Clarity (1), and Significance (1), and the resulting reject recommendation, have been substantially mitigated.
>
> Thank you again for your time and continued effort. Your comments have certainly helped us to significantly improve the readability of our paper.
>
> Sincerely,
>
> Authors

---

### Official Review · Reviewer_Sv4t · 2025-07-01

**Clarity:** 2
**Significance:** 3
**Originality:** 2
**Rating:** 3
**Confidence:** 4

**Summary:**

The paper has proposed a re-selection way of the confident subset identified in the former iteration. To support this, the authors have tried to show some aspects of explaining why this operation is effective. Experimental tests have shown the performance of the selection way. But, in my view, the paper spent a considerable amount of space on meaningless analysis and definitions, resulting in the details of the core algorithm not being provided in the main text. Without so much analysis, the method is still intuitive and effective enough.

**Questions:**

See weaknesses.

**Ethical Concerns:**

["NO or VERY MINOR ethics concerns only"]

**Final Justification:**

I will keep my score

**Limitations:**

Limitation is not clear.

**Quality:**

2

**Strengths And Weaknesses:**

Strengths
1. From the experiments, the proposed algorithm has achieved the SOTA performance.
2. The paper is clear.

Weaknesses
1. Definition of MEE makes little sense. Its mathematical expression is not given.
2. The authors employed around 3 pages to describe a kind of phenomenon that not all mislabeled examples harm the model’s performance equally, and then derived the MEE. But the phenomenon is intuitive. The redundant details did not uncover any significant insight of the phenomenon.
3. In the Methodology section. What is the eventual method? What’s the specific objectives of the method? Why these important details are all in the appendix?

---

> ### Author Rebuttal · Authors · 2025-07-28
>
> Dear Reviewer Sv4t and Area Chair,
>
> Thank you for your thoughtful and constructive feedback on our manuscript. We have carefully revised the manuscript based on your suggestions and have detailed our revisions below.
>
> ---
>
> >**1. On the Justification and Definition of Mislabeled Easy Examples (MEEs)**
>
> Thank you for your feedback. We agree that the high-level notion that not all mislabeled examples are equally harmful can be intuitive. However, we would like to clarify that our core contribution is not merely to restate this intuition. Rather, our goal is to systematically quantify this phenomenon, providing it with a solid empirical and conceptual foundation.
>
> Based on quantitative experiments in Section 2, we extend this high-level notion to uncover a more specific and counter-intuitive finding: *the mislabeled examples learned earliest by the model, which we define as Mislabeled Easy Examples (MEEs), are particularly harmful to generalization.*
>
> This key insight stands in direct opposition to a prevailing assumption in the *Learning with Noisy Labels* field: that early-learned examples are reliable and trustworthy, an assumption that forms the basis of several mainstream methods. Therefore, a detailed analysis in Section 2 is necessary to reveal this potential blind spot in current methodologies and to provide the fundamental motivation for our proposed Early Cutting strategy.
>
> To address your valid concerns about clarity and conciseness, we have undertaken the following significant revisions:
>
> * **Streamlined Analysis:** We have streamlined the presentation of Section 2 to be more direct. We now focus on the most critical evidence supporting our core, counter-intuitive claim (e.g., Figures 1 and 2) and have significantly reduced the descriptive text.
> * **Formal Definition of MEEs:** In response to your request for a more rigorous formulation, we have introduced a precise operational definition in Section 3. Specifically, we have added a clear Definition 1 box, which is directly linked to and cross-referenced with Algorithm 1. As noted below, Algorithm 1 has been moved to the main text.
>
> With these revisions, we aim to make our core conceptual contribution clearer, while conserving space in the main body for the methodological improvements described next.
>
> >**2. On the Clarity and Placement of the Methodology**
>
> We fully agree with your assessment that core algorithmic details should not be relegated to the appendix. We have made the following revisions:
>
> * By streamlining Section 2, we have freed up space in the main body. This has allowed us to move key algorithmic details from Appendix B directly into *Section 3*. We have also included the pseudocode for *Algorithm 1* in the main body to provide a clear, step-by-step description of our final method.
> * In the revised Section 3, we now explicitly state the objectives of our proposed Early Cutting method and explain how the recalibration step is designed to identify and eliminate the MEEs identified in Section 2.
>
> These changes make the methodology section more self-contained and ensure that readers can understand our core technical contribution without referring to the appendix.
>
> >**3. On the Missing Limitations Section**
>
> We have added a *Section 6: Limitations* in the revised manuscript. In this section, we acknowledge the computational overhead of the training process and discuss potential fairness and performance concerns related to the method's application on datasets with class imbalance.
>
> ---
>
> Thank you once again for your valuable feedback.
>
> Best regards,
>
> Authors

---

> > ### Comment · Reviewer_Sv4t · 2025-08-06
> >
> > Thank you for your detailed reply. But I still believe that the paper mainly focuses on testing, discussing, and verifying the issue of 'not all mislabeled examples are equally harmful', without theoretically solving or proving something about the essence of this phenomenon. I personally would not recommend acceptance, but I will not oppose it if the other reviewers are in agreement.

---

> > > ### Author Response · Authors · 2025-08-06
> > >
> > > Dear Reviewer Sv4t,
> > >
> > > Thank you for your continued engagement with our manuscript and for your thoughtful comments.
> > >
> > > We respect and appreciate your observation that our work primarily focuses on empirically identifying and characterizing the phenomenon of disproportionate harm from early-learned mislabeled examples. This accurately summarizes our research paradigm: our paper establishs a clear empirical foundation for this underlying mechanism through experimentation.
> > >
> > > While we have not provided a complete theoretical guarantee, our work follows a classic paradigm in modern machine learning research: Problem Identification → Phenomenon Characterization → Solution. We first discovered, through empirical research, a counter-intuitive pattern, the disproportionate harm of MEEs, which challenges a prevalent assumption in the LNL field. Subsequently, we characterized this phenomenon through both qualitative (feature space analysis) and quantitative (influence functions) methods, and proposed an effective, widely-validated solution: Early Cutting.
> > >
> > > Therefore, we humbly argue that this complete research arc, from identifying an overlooked problem to providing an effective solution, **itself constitutes a core insight and contribution to the essence of this phenomenon**.
> > >
> > > **Toward a Theoretical Grounding**: Your feedback has inspired us to further outline a clearer theoretical framework for our findings.  A full general theory is challenging due to (i) non-convex, data-dependent dynamics in deep nets (which is well know) and (ii) the instance-dependent nature of MEEs (their "easiness" hinges on dataset-specific shortcut cues).
> > >
> > > Despite these challenges, we trust Shortcut Learning [1] could offer a theoretical lens through which to analyze our observations. We can hypothesize that, in the presence of label noise, certain mislabeled examples become MEEs due to the shortcut features they contain, which: 1.  Strongly correlate with their mislabels (e.g., blue backgrounds in "ship→airplane" MEEs); 2.  Are simpler to learn than semantic features (i.e., have lower complexity); 3.  Are captured early in training due to the simplicity bias of gradient descent.
> > >
> > > To formalize this, we propose an analytical framework inspired by the idea of Feature Competition [2, 3], which includes two types of features: Semantic features $f_s$: complex but aligned with the true labels. Shortcut features $f_c$: simple but aligned with the noisy labels.
> > >
> > > Based on this framework, we can clearly analyze the two core phases of learning:
> > > In early stage, due to simplicity bias, gradient descent will preferentially learn the function dominated by $f_c$, leading to the MEE being quickly fitted to its incorrect label. In later stage, to reduce the overall error on the entire dataset, the model is forced to learn the more general semantic features $f_s$. At this point, for an MEE, the prediction dominated by $f_s$ will align with its true label. This, however, creates a conflict with the persistent mislabels in the dataset.
> > > This line of analysis can draw upon recent theoretical work on feature-learning dynamics and the varying speeds at which different features are learned [4, 5, 6].
> > >
> > > ---
> > >
> > > We are grateful for your continued engagement. We hope our detailed response is helpful for your further assessment.
> > >
> > > Best regards,
> > >
> > > Authors
> > >
> > > ---
> > >
> > > Reference:
> > >
> > > [1] Shortcut learning in deep neural networks. Nature Machine Intelligence, 2020.
> > >
> > > [2] Adversarial Examples Are Not Bugs, They Are Features, NeurIPS 2019.
> > >
> > > [3] On the Spectral Bias of Neural Networks, ICML 2019.
> > >
> > > [4] Fine-Grained Analysis of Optimization and Generalization for Overparameterized Two-Layer Neural Networks, ICML 2019.
> > >
> > > [5] SGD on Neural Networks Learns Functions of Increasing Complexity, NeurIPS 2019.
> > >
> > > [6] The Pitfalls of Simplicity Bias in Neural Networks, NeurIPS 2020.

---

### Comment · Reviewer_Q7UE · 2025-08-01
**Emergency Review (Part 1)**

I am providing this emergency review at the behest of the programme committee who asked me to input it as an official comment. Due to space concerns, the review is split in two responses.

**Summary**

The paper addresses the problem of Learning with Noisy Labels (LNL) by challenging the common assumption that all examples learned early in training are reliable. The authors introduce the concept of Mislabeled Easy Examples, a subset of mislabeled samples which is learned quickly and confidently but wrongly, and is shown to be detrimental to model generalisation. To address this, authors propose Early Cutting which acts as a recalibration step for an initial confident set of samples. It leverages a later model to re-evaluate the early-learned confident set. By identifying samples within this set that exhibit high loss, high prediction confidence and low input-gradient norm Early Cutting removes harmful examples. The authors experiment on CIFAR-10/100, CIFAR-N, WebVision and ImageNet-1k and show that their method outperforms other noisy learning techniques

**Strengths and Weaknesses**

**Strengths:**
- The core contribution is novel and important. It improves our understanding of the memorization effect, showing that not all mislabeled samples are "hard" to learn and that the "easy" ones can be damaging. This challenged my notions about sample selection in a positive way.
- The counter-intuitive idea of using a "less reliable" later-stage model to clean up the "more reliable" early-learned set is clever and effective.
- The paper's claims are backed by strong experimentson standard and larger-scale benchmarks with various noise settings. I almost missed the ViT on page 19 of the appendix, it would make sense to more strongly say that the results are not just on ResNets.
- The paper is very well-written, organized and easy to follow. The motivation is clear and the figures are interesting. A note on accessibility though: Figure 4a bottom left is red/green. For protanopic (“colour-blind”) persons, this is hard to read.

**Weaknesses:**
- The explanation for why MEEs are particularly harmful relies on an intuitive, almost anecdotal argument about the model learning “simple but incorrect patterns first”. While the empirical evidence presented is convincing and I believe the argumentation, it would be interesting if this mechanism could be formalised. Although there is a measure of the negative impact of these samples through performance changes, I’d be interested in something like an influence-function-based evaluation of these samples
- The method is developed and tested only on image classification and confidence/CE loss are pretty specific to this task. I’m wondering about transferrability to other tasks like regression, image-to-image and so on. This narrows the applicability somewhat.
- The criteria for identifying MEEs (high loss, high confidence, low gradient norm) are effective but the "gradient stability" criterion feels somewhat ad hoc. While there is a rationale, it is not well-situated within the existing literature in the main text (for example Variance of Gradients, Agarwal 2020 or input-output Jacobian norm, Nowak 2018, Sharpness-aware minimisation etc.). A stronger connection to related work on gradient properties or sharpness-aware minimization would improve the grounding here.
- The way I read it, Early Cutting is presented as an enhancement to Late Stopping. This makes it a bit difficult to disentangle the contribution of Early Cutting from the base method. It would be interesting to see how applicable Early Cutting is when integrated with other types of sample selection methods such as Co-Teaching etc.
- The method introduces new components and hyperparameters such as the thresholds for high loss, high confidence, and low gradient norm, and the "Early Cutting Rate”. Although the fixed percentile-based thresholds are successful and the sensitivity analysis shows good robustness, this still adds a layer of complexity to the training pipeline. A more in-depth discussion on the choice of the default percentile values would be helpful
- The way I understand the method, its effectiveness hinges on using the model from a "later training stage" where it has begun to overfit. While this notion is clear, the practical determination of this epoch without a clean validation is non-trivial in my opinion if we are operating in a label-noise environment as is assumed.

---

> ### Author Response · Authors · 2025-08-04
> **Response to Emergency Reviewer Q7UE (Part 1)**
>
> Dear Reviewer Q7UE and Area Chair,
>
> We sincerely thank you for providing such a prompt, detailed, and constructive emergency review for our paper. We deeply appreciate the time and expertise you have dedicated to our work.
>
> We have comprehensively revised our manuscript based on your feedback. Below, we provide a point-by-point response to each of your questions and concerns.
>
> ---
>
> >**1. Strengthening the Core Concept**
>
> **1.1. Quantitative Analysis of MEE Harm with Influence Functions (in response to Q1)**
>
> Following your excellent suggestion, we have conducted a new quantitative analysis using influence functions to more directly measure the harm caused by MEEs. Specifically, we use influence functions to calculate the impact score of different training samples on a clean, held-out validation set.
>
> This analysis parallels the experiment in Figure 2, comparing three sample categories: (1) MEEs (the first 4,000 mislabeled samples learned during training); (2) Clean Easy Examples (the first 4,000 clean samples learned); and (3) Mislabeled Hard Examples (the last 4,000 mislabeled samples learned). We used the `pytorch_influence_functions` library on CIFAR-10 with 40% instance-dependent noise. The results are as follows:
>
> | Influence Score Statistics | Mean | Median |
> |:---------------------------| :---: | :---: |
> | MEEs                       | **4.96** | **4.48** |
> | Mislabeled Hard Examples   | 3.01 | 3.26 |
> | Clean Easy Examples        | -2.22 | -1.89 |
>
> These results provide direct, quantitative proof of our claim: while all mislabeled examples are harmful (positive influence scores), the average harm of MEEs is over 60% greater than that of Mislabeled Hard Examples (4.96 vs. 3.01). This demonstrates at a microscopic level that MEEs are far more detrimental to the model. In contrast, the negative influence score of Clean Easy Examples (-2.22) confirms their beneficial impact.
>
> The full version of this new analysis will be added to the appendix, and we will reference it in Section 2 to further substantiate our claims about the unique harm of MEEs.
>
> **1.2. Theoretical Grounding of the Gradient Criterion (in response to Q3)**
>
> Thank you for pushing us to better situate our gradient criterion within the literature. We have now added citations and discussion in the Related Work and Methodology (Section 3) sections, connecting our work to *Variance of Gradients (VoG)* [1], *Input-output Jacobian Norm* [2], and *Sharpness-Aware Minimization (SAM)* [3]. In brief:
>
> VoG uses *temporal variance* over training to identify "hard-to-memorize" samples. We use *local input sensitivity* at a later training stage to identify samples that are "*easily and stably mislearned*". The two criteria focus on different dimensions (time vs. input space) and training phases, making them complementary.
> When the model is highly confident in a wrong class, the loss gradient w.r.t. the input, $\|\nabla_x \ell\|$, is dominated by the input-output Jacobian norm. A small $\|\nabla_x \ell\|$ thus indicates that the model's incorrect prediction is insensitive to perturbations in the input neighborhood (a "bad" flat minimum).
> SAM seeks "good" flat minima in the *parameter space* to improve generalization. Our criterion measures sample-wise flatness in the *input space* to identify stability in *incorrect predictions*. The two concepts are therefore complementary.
>
> We have also further clarified that the "low input gradient norm" is not a standalone filter but a *refinement step*. It is only applied to the subset of early-learned samples that already meet the "high loss + confident-and-wrong" criteria, allowing us to distinguish stably mis-memorized MEEs from truly clean but hard examples.
>
> >**2. Exploring Generality**
>
> **2.1. Applying Early Cutting to Loss-Based Methods (in response to Q4)**
>
> To demonstrate that Early Cutting is a general principle, we applied it to a confident set selected by the classic "small-loss trick". On CIFAR-10 with 40% instance-dependent noise, we first selected ~60% of the training data using the small-loss criterion. We then compared the final test accuracy after retraining on this set, with and without an additional Early Cutting step.
>
> | | Small-loss without Early Cutting | Small-loss with Early Cutting |
> | :--- | :---: | :---: |
> | Test Accuracy | 83.55% | 84.30% |
>
> The results show that Early Cutting successfully improves the performance of a loss-based selection method, confirming its modularity. Regarding Co-teaching, we humbly discuss a direct application is inappropriate, as our method's core idea, using a "less reliable" later-stage model to clean up a "more reliable" early-learned set, does not align with the dual-network, peer-teaching dynamic of Co-teaching.

---

> > ### Comment · Reviewer_Q7UE · 2025-08-04
> > **Thank you to the authors**
> >
> > Thank you to the authors for your response. I have read the response and acknowledge the "Mandatory Acknowledgement" as per the NEURIPS 2025 policy.
> >
> > > On influence functions
> >
> > Thank you for turning these experiments around so quickly. I am a bit confused about the direction (you wrote that "all mislabeled examples are harmful (positive influence scores)"). Typically, positive cross-influence describes that the accuracy of a validation example prediction is increased, not decreased. Can you please elaborate what you mean here?
> >
> > > Theoretical Grounding of the Gradient Criterion (in response to Q3)
> >
> > Thank you
> >
> > > Applying Early Cutting to Loss-Based Methods (in response to Q4)
> >
> > Thank you. I understand your concerns about Co-Teaching and will not hold this against you as I admit to also not be deeply familiar with the methods.

---

> ### Author Response · Authors · 2025-08-04
> **Response to Emergency Reviewer Q7UE (Part 2)**
>
> **2.2. Exploring Applicability Across Domains and Tasks (in response to Q2)**
>
> We have added a new appendix section (B.11) to discuss the potential translation of our concepts to other domains, as you suggested. We provide some conceptual examples, in brief:
> In NLP, an MEE could be a text with strong, misleading keywords. For instance, a sarcastic positive review like "Wow, I can't believe how awful the service was" being mislabeled as "Positive". A model might quickly learn this incorrect association due to "Wow" and "can't believe". The Early Cutting criteria could be adapted by using the gradient norm of input word embeddings.
> In regression, an MEE could be a data point where a feature has a strong local linear relationship with the target value, but this relationship is globally spurious. The model would fit this pattern early on. Our criteria (high loss, high confidence in a different value range, and stable gradients) could be conceptually adapted.
>
> >**3. Methodological Details**
>
> **3.1. Precision Analysis of MEE Removal (in response to Q5)**
>
> Thank you for this question. The data is included in the last row of Table 1 (Additional Samples Filtered...). To make this clearer, we present the absolute numbers of "truly mislabeled" vs. "clean but removed (false positives)" for the samples filtered by Early Cutting:
>
> | Number of Samples Filtered by Early Cutting | Truly Mislabeled | Clean but Removed (False Positives) |
> | :--- | :---: | :---: |
> | CIFAR-10 with Sym. 40% noise | 55 | 43 |
> | CIFAR-10 with Asym. 40% noise | 182 | 9 |
> | CIFAR-10 with Pair. 40% noise | 74 | 87 |
> | CIFAR-10 with Ins. 40% noise | 274 | 26 |
>
> **3.2. Further Rationale on Thresholds and the "Later Stage" (in response to Q6 & Weakness 6)**
>
> **On Percentile Thresholds:** Our initial percentile values were determined empirically on a held-out validation set, which is standard practice. However, our core argument is that the method's success is not contingent on these exact values. As the sensitivity analysis in Figure 5 shows, performance is highly stable across a wide range of thresholds.
>
> We agree that the optimal thresholds may correlate with the noise rate, this is a good idea for optimal performance. Given the time constraints in emergency reviewing, we commit to adding an experiment exploring this relationship in the appendix. We note, however, that our fixed thresholds already achieve strong performance across all reported noise rates, although it may not be the best.
>
> **On Determining the "Later Stage":** You raised a point about identifying the early stopping epoch without a clean validation set. We wish to clarify that this is a solvable problem. As detailed in a research [4], the early stopping point in LNL can be accurately determined by monitoring training dynamics on the *noisy validation set* or by observing changes in the model's "fitting performance" on training set, thus making our approach practical in real-world scenarios.
>
> >**4. Revisions to Presentation and Limitations**
>
> Finally, we will adopt all of your presentation suggestions and add a comprehensive Limitations section.
>
> * **Limitations Section:** We have freed up space by streamlining Section 2 and have now included a detailed Limitations section in the main paper. It discusses: (1) scope being limited to visual classification, (2) the empirical nature of the MEE harm justification, (3) computational overhead, and (4) potential risks in extreme noise or imbalanced data scenarios.
> * **Presentation:** We will more explicitly link the **ViT** results in the main text. We will also replace the red/green colors in **Figure 4a** with a colorblind-friendly palette (**#5cb6ea** and **#e5a11c**). We believe that improving the accessibility of every paper is important for promoting fairness and inclusion in the community. We have previously reviewed grayscale images to make the color palette of line graphs friendly to everyone. However, we overlooked the dot graphs such as Figure 4a that you pointed out. We thank you for this thoughtful suggestion.
>
> ---
>
> Once again, we extend our sincerest gratitude for your constructive and prompt review.
>
> Best regards,
>
> Authors
>
> ---
>
> Reference:
>
> [1] Estimating Example Difﬁculty using Variance of Gradients, CVPR 2022.
>
> [2] Sensitivity and Generalization in Neural Networks, ICLR 2018.
>
> [3] Sharpness-aware minimization for efficiently improving generalization, ICLR 2021.
>
> [4] Early Stopping Against Label Noise Without Validation Data, ICLR 2024.

---

> > ### Comment · Reviewer_Q7UE · 2025-08-04
> >
> > Thank you, I am satisfied with these answers. If you would clear up my outstanding point on influence functions above, I would be happy to increase my score.

---

> > > ### Author Response · Authors · 2025-08-04
> > > **Response to Emergency Reviewer Q7UE (Round 2)**
> > >
> > > Dear Reviewer Q7UE,
> > >
> > > Thank you so much for your prompt reply and for acknowledging our responses. We are very pleased to know that we have addressed the vast majority of your concerns. We greatly appreciate your understanding, as we endeavored to be as thorough as possible in the limited time available.
> > >
> > > Regarding your final question on influence functions, in our analysis, we used the following calculation code:
> > > `influence = -sum(torch.sum(g * s).item() for g, s in zip(g_list, s_test_sum))`
> > >
> > > Here, `g` is the gradient of a training sample's loss. The vector `s` is the result of the inverse-Hessian–vector product.
> > >
> > > The logic is that for a beneficial sample, reducing its training loss should also contribute to reducing the test loss. This relationship mathematically results in a positive dot product (`sum(torch.sum(g * s))`). Consequently, due to the leading negative sign in our calculation, the final influence score for a beneficial sample is negative.
> > >
> > > Thank you once again for your valuable feedback.
> > >
> > > Best regards,
> > >
> > > Authors

---

> > > > ### Comment · Reviewer_Q7UE · 2025-08-04
> > > >
> > > > Understood. I was considering the version of influence which uses a probability of correct prediction (where positive influence is a higher probability of a correct prediction), while you seem to be using loss which is an equivalent interpretation. This is fair and thank you for the rapid clarification.
> > > >
> > > > My concerns are now addressed. For the program committee, please note that I would like to increase my score to 5 (Accept).
> > > >
> > > > Thank you to the authors and my fellow emergency reviewers for the productive author-reviewer discussion period.

---

> > > > > ### Author Response · Authors · 2025-08-05
> > > > >
> > > > > Dear Reviewer Q7UE,
> > > > >
> > > > > Thank you for your valuable time and effort in reviewing our manuscript, especially during the tight review period. We sincerely appreciate your meticulous feedback and insightful questions.
> > > > >
> > > > > We are grateful that you have raised your rating to "Accept" following our constructive discussion.
> > > > >
> > > > > Thank you once again for your recognition and support.
> > > > >
> > > > > Best regards,
> > > > >
> > > > > The Authors

---

### Comment · Reviewer_Q7UE · 2025-08-01
**Emergency Review (Part 2)**

**Questions**

- The paper's premise is that MEEs are quite harmful. While the results strongly support this, the explanation remains intuitive. Could you provide a more direct analysis of this harm? For instance, have you considered using cross-influence to quantify the negative impact of the identified MEEs on the model's predictions on a clean validation set?
- Could you elaborate on how you see the core *concept* of MEEs translating to other domains (e.g., NLP, tabular data) or tasks (e.g., regression)? What might a spurious "easy" feature look like in a text classification problem, and how would Early Cutting need to be adapted to handle it?
- The use of low gradient norm wrt input is a key part of the MEE filter algorithm. Can you elaborate on the connection between this criterion and related work as stated above?
- Have you considered applying the Early Cutting principle to a different class of sample selection methods, such as classic loss-based ones like Co-teaching?
- The way I understand it, Table 1 shows the noise rate of the *additional* samples filtered out by the method. For the samples identified specifically as MEEs and removed, what percentage are truly mislabeled and what percentage are clean but were removed (false positives)?
- Can you provide more intuition or theoretical grounding for the percentile values of the thresholds, how they were chosen initially etc? Is there a relationship between the noise rate and the optimal percentile for the loss threshold?

**Limitations**

I’m a bit unhappy with the “NA” for the Limitations. I believe there are limitations that are not catastrophic but should be mentioned. For example limited scope to image classification, informal/empirical justification for MEE harm, computational overhead of keeping two models in memory, overhead due to the reliance of an iterative base method, risks of removing too many clean samples if MEE identification is not good enough (for example in very high or very low-noise regimes).

**Scores**

Quality: 4

Clarity: 4

Significance: 3

Originality: 3

Confidence: 4

**Pre-rebuttal score**: 4 (Borderline Accept)

---

### Comment · Reviewer_99k4 · 2025-08-01
**Review by Reviewer 99k4**

## **Summary**\*

This paper introduces Mislabeled Easy Examples (MEEs), mislabeled samples that models fully learn very early in training but are disproportionately harmful to generalization (esp in comparison to mislabeled hard examples later in training). The authors propose Early Cutting, a method that builds upon existing dynamics-based filters (in particular it is inspired by the analogously named Late Stopping) by adding a re-calibration pass to remove these problematic MEE examples. Early Cutting identifies samples for removal based on three criteria: (i) high loss on the noisy label, (ii) high confidence in the predicted label, and (iii) small input-gradient norm, using fixed percentile thresholds (top 10%, top 20%, bottom 20%). The method is evaluated on CIFAR-10/100, CIFAR-N, WebVision, and ImageNet-1k, showing consistent improvements over recent baselines with comparable computational overhead.

## **Strengths And Weaknesses**\*

**Strengths:**

**Quality**: The paper is well written and provides strong empirical evidence for the MEEs concept, particularly through a controlled "swap-in 4000 mislabeled easy vs. hard examples" experiment, where easy vs hard is determined by when a mislabeled sample is learned for the first time. It convincingly demonstrates that MEEs cause larger accuracy drops than mislabeled hard examples (which are mislabeled but only memorized later in training). The evaluation of Early Cutting is comprehensive, covering synthetic symmetric, asymmetric, pair-flip, and instance-dependent label noise across multiple datasets and comparing to many recent approaches across several datasets. The runtime analysis provides concrete timings showing negligible overhead and similar runtime to other methods with better performance. Ablations in the appendix show that the triple thresholding is beneficial.

**Clarity**: The experiments are well-documented with sufficient detail for reproduction, including optimizers, learning rates, and iterative retention rates in the appendix.

**Significance**: The paper addresses an under-explored aspect of noisy label learning and demonstrates consistent empirical benefits across various datasets and compared to many prior approaches.

**Weaknesses:**

**Writing Quality**: Very minor language issues (very first sentence broken “, while heavily relies”; “These noise are designed”, “was pretrained \[...\] then training”. The paper lacks a dedicated limitations section and doesn't adequately contrast with concurrent work on sample forgetting and gradient-variance criteria. “FkL” is not explained as **First-time k-epoch Learning.**

**Misc.:** The plots and figures are on the small side. I print papers 2x1, so the figures were barely legible. It might be worth moving the CIFAR100 results to the appendix in Figure 2 and 3. And/or the authors could merge the different panels for an individual dataset into a plot  and use different colors for the different mixtures.

## **Quality**\*

❌ 4: excellent

## **Clarity**\*

❌ 3: good

## **Significance**\*

❌ 3: good

## **Originality**\*

❌ 3: good

## **Questions**\*

1. Can you evaluate the method on harder instance-dependent noise for ImageNet (even on a subset) to demonstrate performance under a more realistic large-scale corruption scenarios?
2. How were the thresholds (10, 20, 20\) chosen?
3. Some plots show mean+std: does that mean there were multiple trials? I did not see any information on that mentioned in the paper. How many trials were run? And for which experiments?

## **Limitations**\*

The authors have not adequately addressed limitations. What are potential failure modes of this method? What is left for future work?

## **Rating**\*

❌ 5: Accept: Technically solid paper, with high impact on at least one sub-area of AI or moderate-to-high impact on more than one area of AI, with good-to-excellent evaluation, resources, reproducibility, and no unaddressed ethical considerations.

## **Confidence**\*

❌ 3: You are fairly confident in your assessment. It is possible that you did not understand some parts of the submission or that you are unfamiliar with some pieces of related work. Math/other details were not carefully checked.

## **Ethical Concerns**\*

❌ NO or VERY MINOR ethics concerns only

## **Paper Formatting Concerns**\*

None

## **Code Of Conduct Acknowledgement**\*

❌ Yes

## **Responsible Reviewing Acknowledgement**\*

❌ Yes

---

> ### Author Response · Authors · 2025-08-04
> **Response to Emergency Reviewer 99k4**
>
> Dear Reviewer 99k4 and Area Chair,
>
> We sincerely thank you for your prompt and constructive emergency review, which provides valuable guidance for further improving our manuscript.
> We have formulated a detailed plan for revisions and supplementary experiments based on your questions and suggestions, which we outline below.
>
> ---
>
> >**1. On Evaluating with Instance-Dependent Noise on ImageNet (in response to Q1)**
>
> First, we would like to humbly note the practical reason we did not initially include an experiment with instance-dependent noise on the full ImageNet-1k dataset (only include Symmetric label noise). Standard protocols [1, 2] for generating this noise type on ImageNet-1k require hundreds/thousands of gigabytes of memory; our attempts failed even after allocating 512GB of virtual memory (swap).
>
> However, we fully agree with your point that demonstrating performance under large-scale corruption scenarios is important. While we included the large-scale, real-world WebVision experiment as an evaluation, your suggestion to use a subset of ImageNet is excellent. Given the time constraints of the rebuttal period, we commit to supplementing our paper with this experiment on a large subset of ImageNet (should be much larger than Tiny ImageNet) to validate our method’s effectiveness under these conditions.
>
> >**2. On the Choice of Thresholds (in response to Q2)**
>
> Our initial thresholds were determined empirically on a held-out validation set, which is standard practice in machine learning.
> However, we wish to emphasize that the success of our method is not contingent on these precise values. As shown in the sensitivity analysis in Figure 5, the method’s performance remains highly stable across a wide range of threshold values. This demonstrates the robustness of our combined criteria and explains why a single, fixed set of thresholds can be effective across multiple different datasets and noise settings. We will add this clarification to Section 3 of the revised manuscript.
>
> >**3. On the Number of Experimental Trials (in response to Q3)**
>
> For all experiments reporting `mean±std`, the results are the average and standard deviation over three trials with different random seeds. We have now explicitly added this information to our experimental setup section in the revised paper.
>
> >**4. Addressing Weaknesses and Planned Revisions**
>
> We agree entirely with the weaknesses you identified and have already addressed them in our revised manuscript:
>
> * **Language Issues:** Thank you for pointing out the blemishes in our writing. We have conducted a thorough proofread of the entire manuscript to correct the issues you mentioned and other grammatical errors.
> * **Limitations Section:** We have added a new *Section 6: Limitations*. In this section, we discuss potential failure modes of the method, potential performance and fairness concerns on long-tailed or extremely sparse datasets, and the computational and memory overhead involved.
> * **Figure Readability:** We greatly appreciate your feedback on figure readability. Acknowledging the nine-page limit requires making certain trade-offs. Following your excellent suggestion, we have moved the CIFAR-100 results from Figures 2 and 3 to the appendix. This allowing improvement of the clarity of the main-text figures and frees up space, which we have used to enhance the rigor and completeness of the methodology in Section 3.
>
> ---
>
> Thank you once again for your valuable suggestions for our work.
>
> Best regards,
>
> Authors
>
> ---
>
> Reference:
>
> [1] Part-dependent label noise: Towards instance-dependent label noise, NeurIPS 2020.
>
> [2] Estimating Instance-dependent Bayes-label Transition Matrix using a Deep Neural Network, ICML 2022.

---

> > ### Comment · Reviewer_99k4 · 2025-08-04
> >
> > Thank you for the response! I'm satisfied and will keep my positive score.

---

> > > ### Author Response · Authors · 2025-08-05
> > >
> > > Dear Reviewer 99k4,
> > >
> > > Thank you for your support of our work and for maintaining your positive "Accept" rating after reviewing our response. We are glad that our rebuttal successfully addressed your questions.
> > >
> > > Following your guidance, we have added these details and included a discussion on limitations in the revised version.
> > >
> > > Thank you again for your valuable time and constructive feedback.
> > >
> > > Best regards,
> > >
> > > Authors

---

> > ### Comment · Reviewer_99k4 · 2025-08-04
> >
> > I have read the author rebuttal and considered all raised points., I have engaged in discussions and responded to authors., I have filled in the "Final Justification" text box and updated "Rating" accordingly (before Aug 13) that will become visible to authors once decisions are released., I understand that Area Chairs will be able to flag up Insufficient Reviews during the Reviewer-AC Discussions and shortly after to catch any irresponsible, insufficient or problematic behavior. Area Chairs will be also able to flag up during Metareview grossly irresponsible reviewers (including but not limited to possibly LLM-generated reviews)., I understand my Review and my conduct are subject to Responsible Reviewing initiative, including the desk rejection of my co-authored papers for grossly irresponsible behaviors. https://blog.neurips.cc/2025/05/02/responsible-reviewing-initiative-for-neurips-2025/

---

### Comment · Reviewer_XCQx · 2025-08-02
**Emergency review**

**Summary:**

This paper addresses the challenge of learning with noisy labels by introducing a sample selection approach. The key insight lies in the authors' observation that mislabeled examples which are nevertheless predicted correctly by the model (referred to mislabeled easy examples) can be particularly harmful to training. To mitigate this, the authors propose a filtering strategy named early cutting that leverages the model's predictions at a later stage to identify and discard such samples. Experiments on CIFAR-10, CIFAR-100, WebVision, and ImageNet-1k provide empirical better performance.

**Strengths and Weaknesses**
- Strengths:

1. This paper offers an insightful observation that not all mislabeled examples are equally detrimental to model performance. Specifically, those that are memorized early by the model tend to be more harmful. While this insight is intuitive, it has been largely overlooked in prior sample selection methods based on the small-loss criterion, which assumes that clean samples are learned more quickly by deep networks. However, this assumption fails to account for mislabeled examples with simple patterns that are also likely to incur small losses and thus be incorrectly retained.

2. The controlled experiments are well designed, and the visualizations effectively illustrate that mislabeled yet easy-to-learn examples tend to be memorized due to their misleading features, which align with simple patterns associated with their incorrect labels. Both the quantitative and qualitative analyses are thorough and provide convincing support for the authors’ claims.

3. The writing is clear, and the paper is easy to follow, with a logical progression from the initial observation to the proposed method and its deeper technical details.


- Weaknesses:

1. The proposed method appears somewhat counterintuitive. The authors argue and empirically demonstrate that mislabeled easy examples are learned early by the model and tend to be embedded near the center of their incorrect (noisy) class. Given this observation, it is unclear why, at a later stage of training, the model would assign both high loss and high confidence to these same samples. Further clarification or theoretical justification for this behavior would enhance the contribution. It would be valuable for the authors to visualize the evolution of the feature space, e.g., via t-SNE plots at later stages of training, and to present representative mislabeled easy examples that are effectively filtered out by the proposed early-cutting strategy.

2. Some important hyperparameter details are missing. In particular, it is unclear how the early stopping epoch and the later training stage are determined in practice. Are these values selected via a validation set, heuristics, or fixed schedules? Moreover, are the same settings used across all datasets, or are they dataset-specific? Clarifying these points would improve the reproducibility and practical applicability of the proposed method.

3. Table 5 reports a top-1 accuracy of **72.32%** for CSGN on WebVision, whereas the original CSGN paper [1] reports a significantly higher accuracy of **79.84%**. A similar discrepancy is observed on CIFAR-N. It would be helpful for the authors to clarify the differences in experimental settings that may account for this gap.

4. In Figure 2, although the Clean+(16000:20000] Mislabeled achieves higher test accuracy initially, its performance degrades rapidly with more training epochs. This phenomenon is not clearly explained in the paper. Could the authors provide further analysis or insight into why this subset leads to such rapid overfitting or degradation, despite its early advantage?

[1] Lin, Yexiong, Yu Yao, and Tongliang Liu. "Learning the latent causal structure for modeling label noise." Advances in Neural Information Processing Systems 37 (2024): 120549-120577.

**Quality:** 3:good

**Clarity:** 3:good

**Significance:** 3:good

**Originality:** 3:good

**Rating:**  4: Borderline accept: Technically solid paper where reasons to accept outweigh reasons to reject, e.g., limited evaluation. Please use sparingly.

**Confidence:** 4: You are confident in your assessment, but not absolutely certain. It is unlikely, but not impossible, that you did not understand some parts of the submission or that you are unfamiliar with some pieces of related work.

**Ethical Concerns:** NO or VERY MINOR ethics concerns only

**Paper Formatting Concerns:**
No

**Questions:**

- Is the proposed method effective on Clothing1M?

**Limitations:**

yes

**Code Of Conduct Acknowledgement:** Yes

**Responsible Reviewing Acknowledgement:** Yes

---

> ### Author Response · Authors · 2025-08-04
> **Response to Emergency Reviewer XCQx (Part 1)**
>
> Dear Reviewer XCQx and Area Chair,
>
> We sincerely thank you for providing such a detailed and constructive emergency review of our paper. We have comprehensively revised our manuscript based on your feedback.
>
> ---
>
> >**1. On the "Counter-intuitive" Dynamics of MEEs (in response to Weakness 1)**
>
> You have raised an insightful question: why would an MEE, which is "easy" (low loss) in the early stages, exhibit "high loss" and "high confidence" later in training?
>
> This seemingly counterintuitive behavior is at the core of our Early Cutting method (detailed in section2.2  and section3) and can be explained by the dynamic evolution of the model's representation power as training progresses.
>
> * **Early Stage (Learning Simple Patterns):** At the beginning of training, a deep model prioritizes learning the most salient and easily distinguishable patterns. A typical MEE (e.g., an image of a *ship* on a vast blue ocean, mislabeled as *airplane*) possesses superficial features (like a large blue background resembling the sky) that align well with its incorrect label. Consequently, the early-stage model quickly learns to classify this sample as "airplane," resulting in a low loss, and the sample is thus considered "easy".
>
> * **Later Stage (Learning Semantic Representations):** As training continues, the model is exposed to thousands of true examples of both "airplanes" and "ships". Its feature extractor becomes far more powerful, learning to understand deeper semantic concepts like object contours, textures, and structures. While the model's overall generalization may start to degrade due to the noisy labels, its fundamental **representation power** continues to strengthen (than model in the early-stage).
>
> * **The Shift:** When this more mature, later-stage model re-encounters the same MEE, its powerful classifier now confidently predicts the correct label, "ship". However, the sample's given label is still "airplane". The loss is computed by comparing the model's high-confidence prediction ("ship") against the given incorrect label ("airplane"), which naturally results in a **high loss** value.
>
> To make this dynamic process more intuitive, and in line with your suggestion, we will add a new set of t-SNE visualizations in the appendix to illustrate the feature space evolution at different training stages.
>
> >**2. Clarifications on Experimental Settings and Baseline Results**
>
> **2.1. On the Performance Discrepancy of the CSGN Baseline (in response to Weakness 3)**
>
> You correctly observed a performance gap for the CSGN baseline. This was an intentional result of our experimental design aimed at ensuring fair comparisons.
> To clarify: the higher accuracy reported in the original CSGN paper is the result of their algorithmic pipeline, which includes different backbones (e.g., PreAct ResNet, Inception-ResNet-v2) and advanced modules like semi-supervised learning. In our main experiments (Tables 2-5), we re-implemented all baselines under a unified supervised learning framework. This includes using the same backbone, hyperparameters, and components for all methods to ensure an apples-to-apples comparison. This design ensures that any reported performance differences are attributable to the core noise-handling strategy itself.

---

> ### Author Response · Authors · 2025-08-04
> **Response to Emergency Reviewer XCQx (Part 2)**
>
> **2.2. On the Performance Degradation in Figure 2 (in response to Weakness 4)**
>
> This is a very keen and insightful observation. Analyzing this phenomenon helps to more deeply understand how different types of noisy samples affect learning dynamics.
>
> You correctly noted that the model trained with "Mislabeled Hard Examples" (MHEs) (`Clean + (16000:20000]`) shows a more drastic performance drop after an initial peak. The mechanism is as follows:
>
> * In the early training phase, the model struggles to fit the MHEs because their features are so dissonant with their incorrect labels. It therefore learns primarily from the 30,000 clean samples, building a relatively pure and high-quality simple pattern. This leads to higher initial test accuracy.
> * However, as training progresses, the memorization effect of deep networks begins to dominate. The model starts to forcibly memorize the MHEs it previously ignored. This process severely corrupts the well-established representations, causing a sharp decline in generalization performance.
>
> In contrast, the model trained with MEEs is polluted with incorrect patterns from the very beginning. Its starting point is worse, and while its performance also degrades, it does not exhibit the same "fall from a great height".
>
> We extended the training CIFAR-100 with different label noise to 200 epochs and obtain ed the following results:
>
> | Training Set | Early Stopping Accuracy | Final Accuracy (200 epochs) |
> |---|---|---|
> | Clean + Mislabeled Easy Examples | 56.91% | 54.35% |
> | Clean + Mislabeled Hard Examples | 59.77% | 55.11% |
>
> These extended results confirm our core thesis: although the learning dynamics differ, the final outcome is consistent. Whether at the early stopping point or the end of training, the dataset containing MEEs inflicts more severe and lasting damage on the model's final performance.
>
> >**3. On Practical Details and Reproducibility (in response to Weakness 2)**
>
> To clarify: the "early stopping epoch" and the "later training stage" (which are the same point in time for our method) are determined using a held-out, noisy validation set (10% of the training data, as stated in the main text). This is a standard practice in the LNL field. We monitor the model's accuracy on this validation set and select the epoch just before accuracy begins to degrade due to overfitting. This method for determination is applied consistently across all datasets.
>
> >**4. On the Effectiveness on the Clothing1M Dataset (in response to Questions)**
>
> Our manuscript did not include experiments on Clothing1M because we prioritized CIFAR-N and WebVision as our primary real-world noise benchmarks, complemented by full ImageNet-1k to show scalability.
>
> We would like to humbly note that Clothing1M was not our first choice for evaluation because we found that its validation set also contains significant label noise from machine annotation. This can limit its utility for evaluating SOTA LNL algorithms, as the performance gap between vanilla CE and advanced methods is often less than 5%. We believe that CIFAR-N, with its human-curated noise, currently offers a more stable and reliable benchmark for real-world noise.
>
> Despite these reasons, we agree with your suggestion that Clothing1M can serve as a further validation point for our method's effectiveness. Given the time constraints in emergency reviewing, we commit to adding an experiment on Clothing1M, in the revised manuscript to address your question.
>
> ---
>
> Thank you once again for your valuable and detailed feedback.
>
> Best regards,
>
> Authors

---

> > ### Comment · Reviewer_XCQx · 2025-08-05
> > **Mandatory Acknowledgement by Reviewer XCQx**
> >
> > **Mandatory Acknowledgement:** I have read the author rebuttal and considered all raised points., I have engaged in discussions and responded to authors., I have filled in the "Final Justification" text box and updated "Rating" accordingly (before Aug 13) that will become visible to authors once decisions are released., I understand that Area Chairs will be able to flag up Insufficient Reviews during the Reviewer-AC Discussions and shortly after to catch any irresponsible, insufficient or problematic behavior. Area Chairs will be also able to flag up during Metareview grossly irresponsible reviewers (including but not limited to possibly LLM-generated reviews)., I understand my Review and my conduct are subject to Responsible Reviewing initiative, including the desk rejection of my co-authored papers for grossly irresponsible behaviors. https://blog.neurips.cc/2025/05/02/responsible-reviewing-initiative-for-neurips-2025/

---

> > > ### Comment · Reviewer_XCQx · 2025-08-05
> > >
> > > Thanks for the authors' detailed rebuttal. This rebuttal addresses my major concerns. I will keep the initial positive score.

---

> > > > ### Author Response · Authors · 2025-08-05
> > > >
> > > > Dear Reviewer XCQx,
> > > >
> > > > Thank you for your valuable time and effort in reviewing our manuscript during the tight review period. Your feedback help us clarify our core arguments and refine our experimental details.
> > > >
> > > > We were very pleased to learn that our response addressed your main concerns. We would like to extend our sincere gratitude to you for maintaining your positive evaluation of our work.
> > > >
> > > > Thank you again for your valuable time and constructive feedback.
> > > >
> > > > Best regards,
> > > >
> > > > Authors

---

### Decision · Program_Chairs · 2025-09-17

**Decision:**

Accept (poster)

**Comment:**

The paper studies the hypothesis that mislabeled examples harm model performance to different degrees during training. The authors identify mislabeled easy examples which are correctly predicted by the model during early training are particularly harmful. They propose a method to address this.

This paper has undergone an elaborate review process with a seven reviewers in total, three of which acted as emergency reviewers. Reviewers praised the core concept of the paper that not all mislabeled examples are equally harmful and the significance of this aspect, the experimental evaluation and the clear writing.

On the other hand, reviewers Sv4t & Dz13 pointed out a lack of comparison to baselines but during the discussion period Dz13's concern was addressed (the desired baselines were already in the submitted manuscript). Reviewer 5oqq voiced concerns about a lack of theoretical guarantees but this issue was not seen as a strong enough reason by other reviewers to consider rejection. I therefore recommend acceptance of this paper.